# An advanced cell cycle tag toolbox reveals principles underlying temporal control of structure-selective nucleases

Julia Bittmann[1], Rokas Grigaitis[2], Lorenzo Galanti[1], Silas Amarell[1], Florian Wilfling[3], Joao Matos[2], Boris Pfander[1]*

[1]Max Planck Institute of Biochemistry, DNA Replication and Genome Integrity, Martinsried, Germany; [2]Institute of Biochemistry, Eidgenössische Technische Hochschule, Zürich, Zürich, Switzerland; [3]Max Planck Institute of Biochemistry, Molecular Cell Biology, Martinsried, Germany

**Abstract** Cell cycle tags allow to restrict target protein expression to specific cell cycle phases. Here, we present an advanced toolbox of cell cycle tag constructs in budding yeast with defined and compatible peak expression that allow comparison of protein functionality at different cell cycle phases. We apply this technology to the question of how and when Mus81-Mms4 and Yen1 nucleases act on DNA replication or recombination structures. Restriction of Mus81-Mms4 to M phase but not S phase allows a *wildtype* response to various forms of replication perturbation and DNA damage in S phase, suggesting it acts as a post-replicative resolvase. Moreover, we use cell cycle tags to reinstall cell cycle control to a deregulated version of Yen1, showing that its premature activation interferes with the response to perturbed replication. Curbing resolvase activity and establishing a hierarchy of resolution mechanisms are therefore the principal reasons underlying resolvase cell cycle regulation.

## Introduction

Eukaryotic chromosomes undergo dramatic structural changes during the cell cycle often referred to as the chromosome cycle (*Blow and Tanaka, 2005*). In order to maintain the integrity of genetic information cells need to adjust their DNA repair and genome integrity pathways to the different requirements within this chromosome cycle. Accordingly, many DNA repair enzymes are regulated by transcriptional and post-translational mechanisms or otherwise adjusted to act at specific stages of the cell cycle (*Hustedt and Durocher, 2017*). While our knowledge of regulatory mechanisms has grown over the past years, a question that often arises is whether a certain enzyme or protein has a function, or not, at a specific cell cycle stage.

Answering this question usually involves cell cycle synchronization and sophisticated tools to induce/deplete protein expression at specific time points. Moreover, not all phenotypes can be investigated in single cell cycle experiments. A simple system that promises to overcome these limitations utilizes so called 'cell cycle tags'. The cell cycle tag methodology was initially developed for budding yeast by the Jentsch group (*Karras and Jentsch, 2010*) and expanded by Kolodner and colleagues (*Hombauer et al., 2011*) and Kubota and colleagues (*Johnson et al., 2016*). Cell cycle tagging involves both the replacement of the endogenous promoter of a gene of interest by a cell cycle-regulated promoter as well as the addition of a protein-coding sequence containing a cell cycle-regulated degradation signal (degron), restricting the expression of the fusion protein to a specific phase of the cell cycle. So far, three cell cycle tags have been developed based on the S phase cyclin Clb6, the M phase cyclin Clb2 and the G1 regulator Sic1 (*Karras and Jentsch, 2010*; *Hombauer et al., 2011*; *Johnson et al., 2016*). These tags constrain protein expression to S phase,

*For correspondence:
bpfander@biochem.mpg.de

Competing interests: The authors declare that no competing interests exist.

early M phase and late M to G1 phase, respectively. Single constructs, and also combinations have been used in several studies (*Karras and Jentsch, 2010*; *Hombauer et al., 2011*; *Karras et al., 2013*; *González-Prieto et al., 2013*; *Gonzalez-Huici et al., 2014*; *Menolfi et al., 2015*; *Renaud-Young et al., 2015*; *Johnson et al., 2016*; *Siler et al., 2017*; *Hung et al., 2017*; *Lafuente-Barquero et al., 2017*; *Kahli et al., 2019*; *Lockhart et al., 2019*). Notwithstanding, the current three-construct-system has major limitations: (i) peak expression levels from the three constructs are vastly different (Sic1 > Clb2 > Clb6, compare *Figure 1C*) and (ii) expression levels cannot be adjusted, which can lead to under- or overexpression of the protein of interest. Collectively, these limitations may confound the interpretation of cell cycle tag experiments.

To overcome these limitations, we have set out to generate an advanced toolbox of 46 cell cycle tag constructs with varied expression levels. To achieve these variations in expression, we have used additional promoters/degrons from cyclins Clb5 and Clb1 and introduced chimeric constructs with new promoter/degron combinations (*Figure 1A*). Furthermore, in order to cripple expression from specific promoters, we introduced 5'UTR truncations (*Merrick and Pavitt, 2018*) and upstream out of frame ATGs (*Araujo et al., 2012*; *Yun et al., 2012*; *Dvir et al., 2013*; *Figure 1A*). This construct toolbox will allow comparable and, within a certain range, titratable expression of the protein of interest.

As a proof of principle, we applied the advanced cell cycle tag toolbox to study the regulation of two structure-selective endonucleases (SSEs), Mus81-Mms4 and Yen1. SSEs are involved in many DNA repair pathways and defined by their ability to recognize and cleave branched DNA structures (*Ciccia et al., 2008*; *Schwartz and Heyer, 2011*; *Dehé and Gaillard, 2017*). While required for the corresponding repair mechanisms, it is obvious that cells must also tightly control SSEs, as unscheduled activation of nucleolytic activities might lead to genome instability (*Dehé and Gaillard, 2017*; *Pfander and Matos, 2017*). A number of SSEs have the ability to cleave Holliday junction (HJ) structures and are therefore involved in processing DNA intermediates arising during homologous recombination (HR) and/or as consequence of replication stalling (*Boddy et al., 2001*; *Kaliraman et al., 2001*; *Doe et al., 2002*; *Bastin-Shanower et al., 2003*; *Ciccia et al., 2003*; *Fricke and Brill, 2003*; *Gaillard et al., 2003*; *Ehmsen and Heyer, 2008*; *Ip et al., 2008*; *Jessop and Lichten, 2008*; *Oh et al., 2008*; *Muñoz et al., 2009*; *Rass et al., 2010*; *Wechsler et al., 2011*; *Saugar et al., 2013*; *Wyatt et al., 2013*; *Saugar et al., 2017*). In mitotically dividing budding yeast three HJ-processing SSEs are active – Mus81-Mms4, Yen1 and Slx1 (*Matos and West, 2014*; *Blanco and Matos, 2015*; *Guervilly and Gaillard, 2018*).

The heterodimeric Mus81-Mms4 nuclease is known to undergo cell cycle regulation with the Mms4 subunit becoming phosphorylated in M phase (*Matos et al., 2011*; *Gallo-Fernández et al., 2012*; *Matos et al., 2013*; *Szakal and Branzei, 2013*; *Princz et al., 2017*). This phosphorylation is mediated by the budding yeast cyclin dependent kinase (CDK) Cdk1/Cdc28 and by a complex consisting of two kinases – polo-like kinase Cdc5 and Dbf4-dependent kinase DDK (Cdc7+Dbf4) (*Gallo-Fernández et al., 2012*; *Matos et al., 2013*; *Szakal and Branzei, 2013*; *Princz et al., 2017*), whereby the timing of Cdc5 expression determines the M phase restriction of Mms4 phosphorylation (*Matos et al., 2013*; *Princz et al., 2017*). Cell cycle regulation impinges on Mus81-Mms4 by two mechanisms. While phosphorylation of Mus81-Mms4 directly stimulates its catalytic activity (*Matos et al., 2011*; *Gallo-Fernández et al., 2012*; *Schwartz et al., 2012*; *Matos et al., 2013*), a second layer of cell cycle regulation requires the engagement of Mus81-Mms4 in a phosphorylation-dependent multi-protein complex comprising several scaffold proteins such as Slx4, Dpb11 and Rtt107 (referred to as Mus81 complex hereafter) (*Gritenaite et al., 2014*; *Princz et al., 2015*; *Princz et al., 2017*). The Mus81 complex forms exclusively during M phase and is likely involved in targeting Mus81-Mms4 to its substrates or controlling its action by other means (*Gritenaite et al., 2014*; *Princz et al., 2017*). Intriguingly, phosphorylation of Mus81-Mms4 and formation of the Mus81 complex displays features commonly associated with switch-like activation (positive feedback, multi-site phosphorylation), suggesting that with the transition to M phase cells might enter a state of increased Mus81-Mms4 function (*Pfander and Matos, 2017*; *Princz et al., 2017*) (note that function in vivo will not only be determined by enzymatic activity, but also by targeting of the enzyme to its substrate, etc). Notably, however, *mus81* mutant phenotypes suggest that the main function of Mus81-Mms4 can be attributed to the response to replication perturbation (*Xiao et al., 1998*; *Interthal and Heyer, 2000*; *Boddy et al., 2001*; *Mullen et al., 2001*; *Doe et al., 2002*; *Bastin-Shanower et al., 2003*; *Kai et al., 2005*). This raises the question, whether (i) Mus81-Mms4 may be

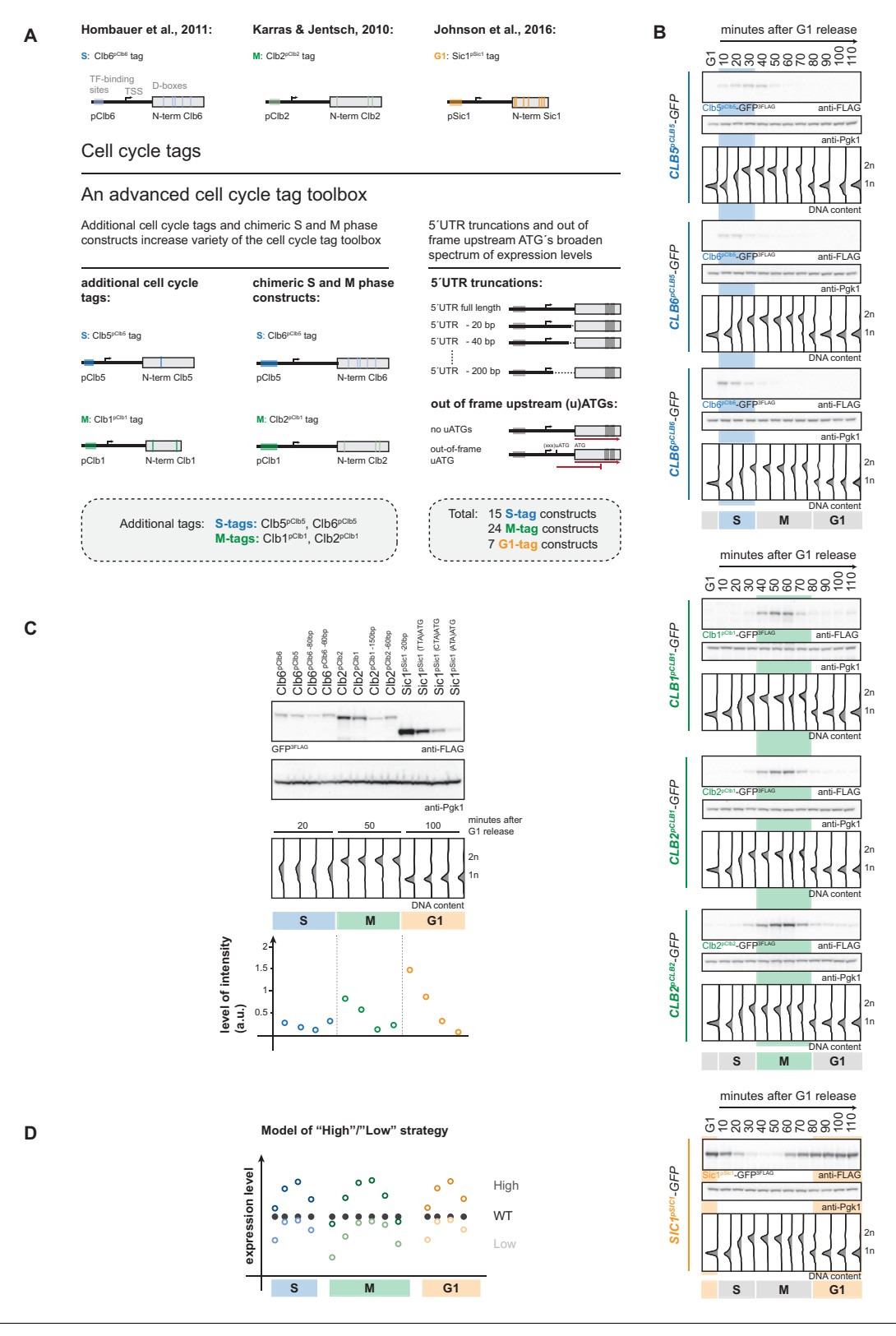

**Figure 1.** An advanced toolbox of cell cycle tag constructs. (**A**) Schematic representation of the applied strategies for improved cell cycle tag methodology. Upper panel: conventional cell cycle tag methodology was limited by only one construct for each cell cycle phase. Lower panel: the advanced cell cycle tag toolbox was expanded to 46 constructs. Therefore, we used new promoters and degrons from Clb5 and Clb1, chimeric promoter-degron combinations and protein expression was crippled using 5′UTR truncations and upstream out of frame ATGs. Vertical bars indicate

*Figure 1 continued on next page*

*Figure 1 continued*

location of cell cycle regulatory elements in the promoter and N-terminal degron sequence (see *Figure 1—figure supplement 1* for detailed description of the tagging procedure). (B) New cell cycle tag constructs allow cell cycle-restricted expression of GFP at varied peak expression levels. Anti-FLAG westerns of cells expressing Clb5$^{pClb5}$-, Clb6$^{pClb5}$-, Clb6$^{pClb6}$-, Clb1$^{pClb1}$-, Clb2$^{pClb1}$-, Clb2$^{pClb2}$-3FLAG-tagged versions of GFP after G1 arrest with α-factor and synchronous release through the cell cycle up to the next G1 phase. Pgk1 western was used as control and DNA content measurements indicate cell cycle progression at the individual time points below (see *Figure 1—figure supplement 2* for G1 release experiments of the corresponding 5′UTR truncations and for constructs containing upstream out of frame ATGs). (C) New promoters and degrons, chimeric promoter-degron combinations, 5′UTR truncations and upstream out of frame ATGs allow a broad spectrum of peak expression levels of cell cycle-restricted GFP. Western blot analysis of peak expression levels of cell cycle-tagged 3FLAG-GFP variants at indicated time points after G1 release (20 min = S phase, 50 min = M phase, 100 min = G1 phase). DNA content measurements below indicate cell cycle progression. Graph: peak expression levels were quantified using Image-J and signals of the individual constructs were divided by the corresponding Pgk1 sample to normalize to overall protein levels (see *Figure 1—figure supplement 3* for an overview of all cell cycle-tagged GFP versions in cells arrested in the corresponding cell cycle phase). (D) Schematic representation of the suggested cell cycle tag strategy using two sets of constructs with matching 'low' and 'high' peak expression levels. 'Low' expressing constructs (light colours) are chosen by matching peak expression levels similar to the endogenous protein but will show underexpression at cell cycle phase transitions. 'High' expressing tags generally show higher expression compared to the wildtype protein with the advantage of broader timeframes of action (timeframes in which protein levels are similar or higher than endogenous protein levels).

The online version of this article includes the following figure supplement(s) for figure 1:

**Figure supplement 1.** Schematic representation of the cell cycle tagging procedure.
**Figure supplement 2.** 5′UTR truncations and upstream out of frame ATGs do not interfere with cell cycle restriction.
**Figure supplement 3.** 5′UTR truncations and upstream out of frame ATGs cripple peak expression levels of cell cycle tag constructs.
**Figure supplement 4.** Clb6$^{pClb6}$- and Clb2$^{pClb1}$-tag induce similar peak expression levels for several cell cycle-tagged proteins.

acting in S phase directly on stalled replication forks or repair intermediates, despite a non-matching temporal regulation, or whether (ii) Mus81-Mms4 acts in M phase as post-replicative resolvase.

A second SSE with the propensity to cleave HJ structures is called Yen1 (*Ip et al., 2008*; *Blanco et al., 2010*). Yen1 is also tightly cell cycle-controlled and becomes dephosphorylated in late M phase, specifically at the metaphase-to-anaphase transition, when CDK becomes inactivated and phosphorylation marks on Yen1 are removed by Cdc14 (*Kosugi et al., 2009*; *Matos et al., 2011*; *Blanco et al., 2014*; *Eissler et al., 2014*; *García-Luis et al., 2014*). Yen1 regulation consists of several layers and involves phosphorylation-dependent inhibition of its catalytic activity as well as phosphorylation-dependent regulation of its sub-cellular localization (*Matos et al., 2011*; *Blanco et al., 2014*; *Eissler et al., 2014*; *García-Luis et al., 2014*). Furthermore, at the G1/S transition a degradation mechanism is in place to clear Yen1 from chromatin (*Talhaoui et al., 2018*). Altogether, a picture emerges whereby Yen1 is inhibited by CDK phosphorylation and becomes stimulated or activated from late M phase to the end of G1 (*Blanco et al., 2014*; *Eissler et al., 2014*; *García-Luis et al., 2014*). The temporal windows of high Mus81-Mms4 activity and high Yen1 activity therefore appear non-overlapping (*Matos et al., 2011*). Experimental removal of the inhibitory phosphorylation sites on Yen1 generated an allele (*YEN1-ON*), where Yen1 was found to be uncoupled from cell cycle regulation and constitutively active (*Matos et al., 2013*; *Blanco et al., 2014*). This allele allowed to study the consequences of unrestricted activation of an SSE and showed that ectopic nuclease activity has adverse consequences in presence of replication stalling agents, suggesting that unscheduled cleavage of replication intermediates by this SSE interferes with the response to replication stalling (*Blanco et al., 2014*).

The control of SSEs in human cells involves additional features, such as the presence of two mutually exclusive MUS81 regulators (called EME1 and EME2), but the principal mechanisms of control appear to be evolutionary conserved (*Matos et al., 2011*; *Wyatt et al., 2013*; *Chan and West, 2014*; *Duda et al., 2016*), suggesting that cell cycle control of SSEs is an intrinsic necessity. In this study, we take advantage of the genetic tractability of budding yeast to expand, improve and apply cell cycle tag technology with an advanced toolbox of cell cycle tag constructs to investigate the relevance of Mus81-Mms4 and Yen1 cell cycle regulation. We show that several survival and genome instability phenotypes induced by chronic or acute exposure to DNA damaging agents or other genotoxic agents are rescued by M phase restricted Mus81-Mms4, but not by a version that is confined to S phase. This suggests that for the conditions tested, the essential function of Mus81-Mms4 is as a post-replicative resolvase. Yen1 can compensate for this function, if present in constitutively active form in early M phase. We also employ cell cycle tags to reintroduce cell cycle regulation and find that premature activation of Yen1 in S phase, but also in early M phase interferes with the response

to replication fork stalling, suggesting that a temporal hierarchy of HJ-cleaving nucleases is required for optimal DNA repair.

## Results

### An advanced toolbox of cell cycle tags

When we started this study, three cell cycle tag constructs were available to restrict target protein expression to G1, S or M phase. The S-tag uses promoter and N-terminal degron (aa 1–195) of the S phase cyclin Clb6 and restricts expression to S phase (*Figure 1A*; *Hombauer et al., 2011*). The M-tag (originally referred to as G2-tag; *Karras and Jentsch, 2010*) uses promoter and N-terminal degron sequence (aa 1–180) of the M phase cyclin Clb2 and restricts expression to M phase (*Figure 1A*; *Karras and Jentsch, 2010*). The G1-tag uses promoter and N-terminal degron of the G1 regulator Sic1 (aa 1–105) and restricts protein expression to G1 (*Figure 1A*; *Johnson et al., 2016*). In order to overcome the limitations of these specific constructs and allow for modulation of expression levels we generated a toolbox of 46 cell cycle tag constructs. Specifically, we used a three-pronged approach to create constructs that at the same time allow varied peak expression levels and retained cell cycle restriction (*Figure 1A*): (i) we used additional promoters and N-terminal degron sequences from S phase cyclin Clb5 (aa 1–202) and from M phase cyclin Clb1 (aa 1–120); (ii) we generated chimeric S-tag and M-tag constructs (containing Clb5 promoter and Clb6 degron (Clb6$^{pClb5}$-tag) or Clb1 promoter and Clb2 degron (Clb2$^{pClb1}$-tag), respectively); (iii) in order to cripple expression from some promoters, we either truncated 5'UTRs or introduced out-of-frame ATGs, which have been shown to reduce protein expression levels by reduced mRNA stability and reduced translation rates, respectively (*Yun et al., 2012*; *Araujo et al., 2012*; *Dvir et al., 2013*; *Merrick and Pavitt, 2018*).

We constructed 46 plasmids based on the pYM-N vector series (*Janke et al., 2004*) that allow the introduction of all variants of cell cycle tags by a well-established recombination-based strategy using a single pair of oligonucleotide primers and the natNT2 resistance cassette (*Figure 1—figure supplement 1*, see supplementary methods for detailed protocol). As test substrate, we subjected the GFP ORF, which was integrated in the yeast genome, to the cell cycle tagging approach and the resulting yeast strains were verified for genomic integration and expression. Next, we tested whether all constructs restricted protein expression to the desired cell cycle phases. Therefore, we arrested cells in G1 using α-factor, synchronously released them into the cell cycle and followed them to the next G1. Notably, all constructs restricted GFP-expression to the target cell cycle phase (*Figure 1B*, *Figure 1—figure supplement 2*). When comparing the different S-tag constructs, we noticed that constructs containing the Clb6 degron sequence imposed a much sharper restriction of expression to S phase consistent with the differential regulation of Clb5 and Clb6 (*Figure 1B*; *Kühne and Linder, 1993*; *Schwob and Nasmyth, 1993*; *Jackson et al., 2006*), suggesting that these constructs should be the preferred choice for an S-tag experiment.

We then took S, M and G1 phase samples (20, 50 and 100 min, respectively) from our cell cycle release experiments in order to measure peak expression levels for the individual constructs. This analysis showed that within different G1-, S- and M-tag constructs expression varied by up to 10-fold, respectively (*Figure 1C*, *Figure 1—figure supplement 3*). Notably, none of the S-tag constructs tested gave peak expression levels in the same range as those of the previously used Clb2 M-tag and Sic1 G1-tag constructs (*Figure 1C*, *Figure 1—figure supplement 3*), emphasizing the need for new M- and G1-tag constructs. Satisfyingly, we found that M-tag constructs containing the pClb1 promoter or the 5'UTR-truncated pClb2 promoter showed much weaker peak expression levels and tighter temporal restriction of expression at the same time (*Figure 1B–C*, *Figure 1—figure supplements 2* and *3*). Similarly, for the G1-tag we found that upstream out-of-frame ATGs reduced protein expression from pSic1 constructs and also led to tighter temporal restriction of expression (*Figure 1B–C*, *Figure 1—figure supplements 2* and *3*).

Therefore, these constructs from our advanced cell cycle tag toolbox will allow to titrate peak expression levels within a certain range and offer at the same time superior restriction of target protein expression to the cell cycle phase of interest. We also note that by introducing cell cycle-restricted expression, one will usually change expression of the protein of interest from a continuous, often constant expression regime to a dynamically, cell cycle phase-restricted expression regime

(*Figure 1D*). Due to this dynamic expression it may sometimes be difficult to directly compare cell cycle-tagged constructs with endogenous proteins, as well as the arising phenotypes. To mitigate this problem, we therefore developed a strategy where we conducted experiments with two sets of G1-, S- and M-tag constructs (*Figure 1D*). The first set (called 'low' hereafter) can be chosen to yield peak expression similar to the protein of interest but may show 'under-expression' at cell cycle transitions (*Figure 1D*). The second set (called 'high' hereafter) can be chosen to yield peak expression higher to the protein of interest (overexpression) but will avoid under-expression at cell cycle transitions (*Figure 1D*). Most importantly, with the presented toolbox it will be possible to use constructs, which give highly similar peak expression levels in different cell cycle phases and the respective strains are therefore phenotypically comparable. Protein expression should be tested for any new cell cycle-tagged protein, even though we observed similar trends for different proteins tested. For example, we found that S phase levels of Clb6$^{pClb6}$-tagged proteins were very similar to levels of Clb2$^{pClb1}$-tagged proteins in five (Xrs2, Rad52, Fun30, Sgs1, Yen1-ON) (*Figure 1—figure supplement 4*, *Figure 5—figure supplement 2*) out of six cases (Mus81-Mms4 being the exception). Overall, our advanced cell cycle tag toolbox therefore offers titratable expression levels, which allows for the first time a direct comparison of phenotypes arising from cell cycle restriction of a protein of interest to G1, S or M phase.

## Cell cycle-restricted expression of Mus81-Mms4

To showcase the cell cycle tag toolbox, we applied it to Mus81-Mms4. Deletion of *MUS81* or *MMS4* causes phenotypes that imply Mus81-Mms4 in the cellular response to replication fork stalling (*Xiao et al., 1998*; *Interthal and Heyer, 2000*; *Boddy et al., 2001*; *Mullen et al., 2001*; *Doe et al., 2002*; *Bastin-Shanower et al., 2003*; *Kai et al., 2005*; *Saugar et al., 2013*). In contrast, Mus81-Mms4 function is specifically upregulated once cells enter M phase (*Matos et al., 2011*; *Gallo-Fernández et al., 2012*; *Matos et al., 2013*; *Saugar et al., 2013*; *Szakal and Branzei, 2013*; *Gritenaite et al., 2014*; *Princz et al., 2017*). We therefore decided to employ our toolbox to discriminate between potential S phase- and M phase-specific functions of Mus81-Mms4. In addition to the strategy outlined in *Figure 1*, we constructed cell cycle tags for both subunits of the Mus81-Mms4 heterodimer, as we reasoned that this would result in even tighter cell cycle restriction of the complex. Specifically, we found that Clb6$^{pClb6\ -80bp}$-tagged and Clb2$^{pClb1\ -150bp}$-tagged versions of Mus81-Mms4 restricted Mus81-Mms4 expression to S and M phase and resulted in very similar peak expression levels between 0.9 and 1.2-fold of the endogenous proteins (*Figure 2A*, *Figure 2—figure supplement 1A,C*). We therefore refer to these versions as S$^{low}$-Mus81-Mms4 (Clb6$^{pClb6\ -80bp}$-tag) and M$^{low}$-Mus81-Mms4 (Clb2$^{pClb1\ -150bp}$-tag), respectively. While peak expression levels are comparable to endogenous Mus81-Mms4, we observed reduced expression levels at cell cycle transitions. For example, we observed that expression of M$^{low}$-Mus81-Mms4 was below endogenous levels in early M phase (see *Figure 2B*, 37.5 and 45 min time points). The same trend was also observed for the nuclear fraction of Mus81-Mms4 (*Figure 2B*, right panel, *Figure 2—figure supplement 2*). Consistently, the window of time during which we could observe the M phase specific, hyperphosphorylated form of Mms4 was shorter for M$^{low}$-Mus81-Mms4 compared to endogenous Mus81-Mms4 (*Figure 2C*). Taken together, M$^{low}$-Mus81-Mms4 peak expression is comparable to endogenous Mus81-Mms4, but expression and hyperphosphorylation appears reduced in early and late M phase. To complement S$^{low}$-Mus81-Mms4 and M$^{low}$-Mus81-Mms4, we therefore also used the Clb5$^{pClb6}$-tagged S phase-restricted S$^{high}$-Mus81-Mms4, as well as Clb2$^{pClb1}$-tagged M phase-restricted M$^{high}$-Mus81-Mms4 (*Figure 2A*). Comparison of peak expression levels in S and M phase suggests that S$^{high}$-Mus81-Mms4 and M$^{high}$-Mus81-Mms4 are expressed to very similar levels, but 2 to 5-fold overexpressed compared to endogenous Mus81-Mms4 (*Figure 2—figure supplement 1B, C*). Both constructs showed expected restriction of expression to S and M phase (*Figure 2A*) and when we compared M$^{high}$-Mus81-Mms4 to M$^{low}$-Mus81-Mms4, we noticed that the M$^{high}$-Mus81-Mms4 did neither show underexpression in early M phase (*Figure 2B*), nor a shortened window of M phase-specific Mms4 phosphorylation (*Figure 2C*). Therefore, S$^{high}$-Mus81-Mms4 and M$^{high}$-Mus81-Mms4 constructs are expressed to similar levels, avoid under-expression at cell cycle transitions, but show overexpression compared to endogenous expression levels. The two sets of S- and M-tag constructs are therefore complementary and enable us to follow the high/low expression strategy outlined in *Figure 1D* to investigate Mus81-Mms4 phenotypes.

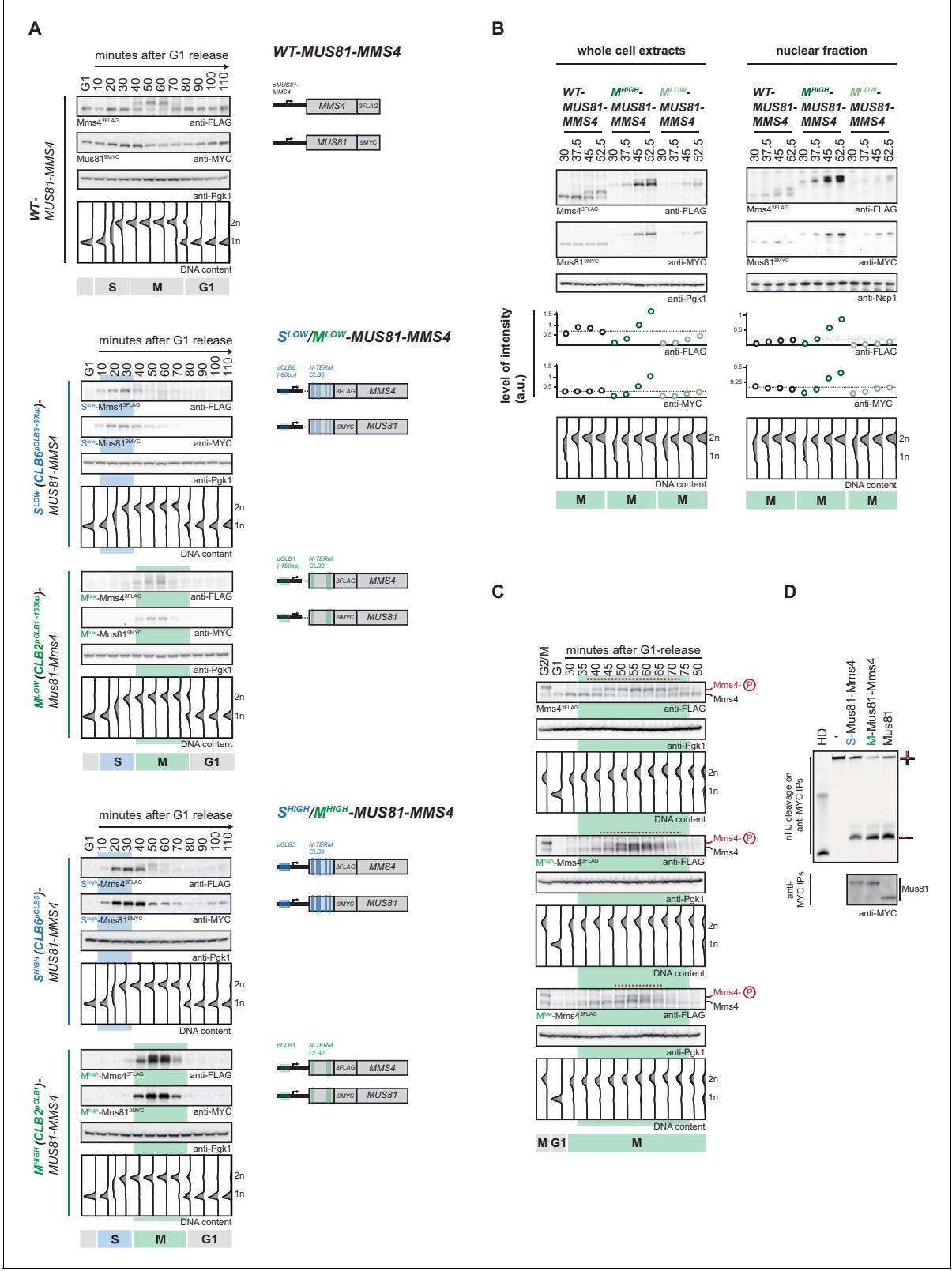

**Figure 2.** Cell cycle-restricted expression of Mus81-Mms4. (**A**) Restriction of Mus81-Mms4 expression to S or M phase of matched pairs of 'low' and 'high' expressing cell cycle tag constructs. (Left) Western blot and DNA content analysis of strains expressing WT, S phase-restricted (S$^{low}$ (Clb6$^{pClb6 -80bp}$)-/S$^{high}$ (Clb6$^{pClb5}$)-Mus81-Mms4) and M phase-restricted (M$^{low}$ (Clb2$^{pClb1 -150bp}$)-/M$^{high}$ (Clb2$^{pClb1}$)-Mus81-Mms4) alleles of Mus81 (9MYC-tagged)-Mms4 (3FLAG-tagged) during a single cell cycle as in *Figure 1D* (see *Figure 2—figure supplement 1* for quantification of peak expression levels of

*Figure 2 continued on next page*

*Figure 2 continued*

the S^low^-/M^low^-Mus81-Mms4 and S^high^-/M^high^-Mus81-Mms4 constructs). (Right) Schematic representation of WT, S (S^low^-/S^high^-Mus81-Mms4) and M phase (M^low^-/M^high^-Mus81-Mms4) restricted Mus81-Mms4 constructs. Blue and green bars indicate location of cell cycle regulatory elements in the promoter and N-terminal degron sequence. (B) Different constructs ('high' and 'low' peak expression levels) of Mus81-Mms4 lead to underexpression of the M^low^-Mus81-Mms4 construct or to overexpression of the M^high^-Mus81-Mms4 construct in early M phase and similar trends are seen in the nuclear fraction. Western blot analysis of protein levels in whole cell extracts (left panel) and after nuclei separation (right panel) at indicated time points after a G1-release (early M phase; see DNA content profile depicted at the bottom). While immediately with entry into M phase the M^high^-Mus81-Mms4 construct reaches similar or higher protein levels than endogenous Mus81-Mms4, the M^low^-Mus81-Mms4 construct shows a 10–15 min delay in reaching comparable expression levels and this holds true for both, whole cell extracts and the nuclear fraction. Expression levels were quantified using Image-J and signals of the individual time points were divided by the corresponding Pgk1 (whole cell extracts) or Nsp1 (nuclear fraction) signal to normalize to overall protein levels (graphs below contain normalized values for every construct). (see *Figure 2—figure supplement 2* for control western blots of the nuclear fractionation) (C) Different constructs ('high' and 'low' peak expression levels) of Mus81-Mms4 lead to different windows of Mus81-Mms4 phosphorylation in M^low^-Mus81-Mms4 and M^high^-Mus81-Mms4. Western blot analysis of the phosphorylation states of Mms4 at indicated time points after a G1-release (M phase; see DNA content profile depicted below the western blots). While M^high^-Mus81-Mms4 shows a similar timeframe of Mms4 phosphorylation to endogenous Mus81-Mms4, M^low^-Mus81-Mms4 is phosphorylated and stimulated during a shortened window of time only (compare red lines above the Mms4-3FLAG western blots: 15–20 min of phosphorylation in M^low^-Mus81-Mms4 compared to 30 min in M^high^-Mus81-Mms4 and 30–35 min in Mus81-Mms4). (D) N-terminal tagging does not alter Mus81-Mms4 activity. Resolution assay using a nicked HJ (nHJ) substrate and immunopurified Mus81-Mms4, S-Mus81-Mms4 and M-Mus81-Mms4 (note that WT and cell cycle-tagged proteins were expressed from pGal1-10 promoter). Myc-tagged Mus81-Mms4 was purified from cycling cells, dephosphorylated using λ-Phosphatase and incubated with the nHJ substrate for 2 hr. Upper panel: nHJ cleavage assay with heat DNA substrate (HD) as control. Lower panel: western blot analysis of Mus81-9MYC IP after nHJ cleavage assay (see *Figure 2—figure supplement 3* for a western blot analysis of Mms4 dephosphorylation by λ-Phosphatase).

The online version of this article includes the following figure supplement(s) for figure 2:

**Figure supplement 1.** Analysis of 'low' and 'high' S- and M-tagged Mus81-Mms4 peak expression levels.
**Figure supplement 2.** Nuclear/cytoplasmic fractionation.
**Figure supplement 3.** λ-Phosphatase treatment leads to efficient Mms4 dephosphorylation of WT, S- and M-Mus81-Mms4 used for activity assays.

Lastly, we also ensured that N-terminal tagging of Mus81 or Mms4 did not lead to inactivation of the Mus81-Mms4 enzymatic activity. To this end we obtained Mus81-Mms4, Clb6 (S-)-tagged Mus81-Mms4 and Clb2 (M-)-tagged Mus81-Mms4 by immuno-purification, phosphatase treated the protein to exclude cell cycle-dependent stimulatory effects and found that all versions showed nuclease activity towards a nicked Holliday junction (nHJ) model substrate (*Figure 2D*, *Figure 2—figure supplement 3*).

## Mus81-Mms4 restricted to M phase, but not S phase is sufficient for the response to genotoxic insults

To reveal phenotypes of Mus81-Mms4 cell cycle restriction, we first tested cell viability upon chronic exposure to replication stalling chemicals (MMS, CPT and HU). *mus81Δ* cells were hypersensitive to MMS and CPT and showed reduced growth on HU containing medium (*Figure 3A–B*; *Xiao et al., 1998*; *Interthal and Heyer, 2000*; *Mullen et al., 2001*; *Bastin-Shanower et al., 2003*; *Saugar et al., 2013*). Restricting Mus81-Mms4 to S phase also gave a severe hypersensitivity to MMS and CPT: S^low^-Mus81-Mms4 expressing cells showed similar phenotypes as *mus81Δ* (*Figure 3A*, *Figure 3—figure supplement 1A*), while S^high^-Mus81-Mms4 also showed pronounced hypersensitivity, but compared to *mus81Δ* (and S^low^-Mus81-Mms4) the phenotype was less severe (*Figure 3B*). In contrast, restricting Mus81-Mms4 to M phase showed very little phenotype. Specifically, M^high^-Mus81-Mms4 cells did not show any hypersensitivity (*Figure 3B*), while M^low^-Mus81-Mms4 cells showed a small but detectable growth defect in the presence of high doses of MMS or CPT (*Figure 3A*). Collectively, these data suggest that Mus81-Mms4 would exhibit its dominant function in the response to genotoxic agents during M phase and not during S phase.

We interpret the slight phenotypic differences between S^high^-Mus81-Mms4 and S^low^-Mus81-Mms4, as well as between M^high^-Mus81-Mms4 and M^low^-Mus81-Mms4 (*Figure 3A–B*), to arise from leaking of S^high^-Mus81-Mms4 into M phase (*Figure 2A*) as well as from underexpression of M^low^-Mus81-Mms4 during early and late M phase (*Figure 2B–C*), respectively. This interpretation is supported by the facts that (i) the residual viability of S^high^-Mus81-Mms4 depends on M phase-specific phosphorylation events (*Figure 3—figure supplement 1B–C,M* phase specific phosphorylation is abolished by the *mms4-14A* mutant; *Matos et al., 2011*) and (ii) the MMS and CPT sensitivity of

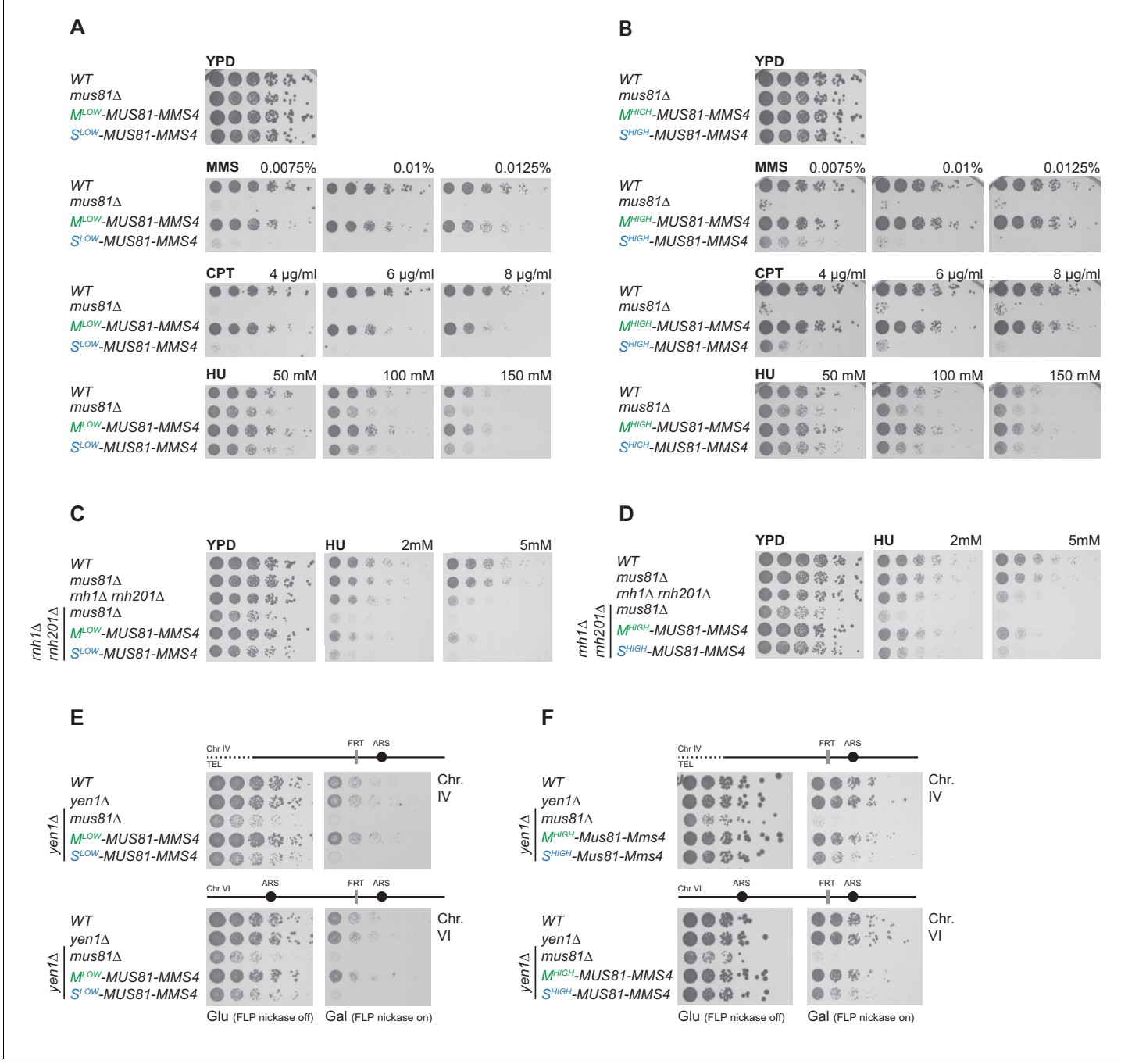

**Figure 3.** Mus81-Mms4 restricted to M phase, but not S phase is sufficient for the response to genotoxic insults. (**A/B**) M phase-restricted Mus81-Mms4 is sufficient to confer viability to replication fork stalling drugs. Viability of cells with M$^{low}$ (Clb2$^{pClb1\ -150bp}$)-Mus81-Mms4/S$^{low}$ (Clb6$^{pClb6\ -80bp}$)-Mus81-Mms4 constructs at low peak expression levels (**A**) or M$^{high}$ (Clb2$^{pClb1}$)-Mus81-Mms4/S$^{high}$ (Clb6$^{pClb5}$)-Mus81-Mms4 constructs at high peak expression levels (**B**) is compared to that of *WT* and *mus81Δ* cells. Strains were plated in 5-fold serial dilutions on YPD plates containing the indicated amounts of MMS, CPT or HU and incubated at 30˚C for 2 days. (**C/D**) Mitotic function of Mus81-Mms4 is sufficient to confer viability upon induction of RNA-DNA-hybrids in the absence of RNAse H enzymes and mild replication stress (HU). Cell cycle-tagged versions of Mus81-Mms4 were integrated in the *rnh1Δ rnh201Δ* background. Strains were spotted in 5-fold serial dilutions on YPD containing indicated concentrations of HU and incubated at 30˚C for 2 days. (**C**) Spotting containing M$^{low}$-Mus81-Mms4/S$^{low}$-Mus81-Mms4 cells. (**D**) Spotting of M$^{high}$-Mus81-Mms4/S$^{high}$-Mus81-Mms4 cells. (**E/F**) Repair of Flp-nickase induced DNA lesions requires the M phase function of Mus81-Mms4. Galactose-induced DNA nicking is presumed to be followed by replication run-off in S phase to form single-ended DSBs and repair by BIR (*Nielsen et al., 2009*; *Mayle et al., 2015*). Location of the corresponding FRT sites on chromosome IV and VI are indicated relative to replication origins. Cells were spotted in 5-fold serial dilutions in presence of glucose or

*Figure 3 continued on next page*

*Figure 3 continued*

galactose (FLP-induction) and incubated at 30°C for 2 days. (**E**) Spottings of M$^{low}$-Mus81-Mms4/S$^{low}$-Mus81-Mms4 cells. (**F**) Spottings of M$^{high}$-Mus81-Mms4/S$^{high}$-Mus81-Mms4 cells.

The online version of this article includes the following figure supplement(s) for figure 3:

**Figure supplement 1.** Residual S$^{high}$-Mus81-Mms4 function in response to genotoxic agents is explained by insufficient restriction to S phase; slight M$^{low}$-Mus81-Mms4 defect in response to genotoxic agents is due to underexpression during M phase, respectively.

M$^{low}$-Mus81-Mms4 cannot be rescued by an additional copy of S$^{low}$-Mus81-Mms4 (*Figure 3—figure supplement 1D–E*).

We next tested whether endogenous replication stress would require Mus81-Mms4 during M phase as well or whether S phase Mus81-Mms4 could play a role in this context. Absence of RNaseH1 and RNaseH2 generates replication stress due to defects in the removal of RNA-DNA-hybrids and defects in ribonucleotide excision repair (*Sollier and Cimprich, 2015*; *Hamperl and Cimprich, 2016*). Mus81 orthologs were first implicated in the replication stress caused by RNaseH-deficiency because of the synthetic growth phenotype of a *mus81Δ rnh1Δ rnh201Δ* mutant in *S. pombe* (*Zhao et al., 2018*). We observe a similar synthetic growth phenotype in the corresponding budding yeast *mus81Δ rnh1Δ rnh201Δ* mutant, which was further aggravated by addition of HU in low concentrations (*Figure 3C–D*). Notably, presence of M phase-restricted versions of Mus81-Mms4 was able to rescue these phenotypes back to levels of the *rnh1Δ rnh201Δ* strain, while S phase-restricted versions of Mus81-Mms4 were unable to do so (*Figure 3C–D*). Furthermore, we studied the response to a site-directed protein-bound single strand break induced by a step-arrest mutant of the Flp recombinase (Flp nickase; *Nielsen et al., 2009*; *Mayle et al., 2015*). Previous studies have suggested that single strand breaks generated in the Flp nickase system would lead to replication run-off (replication fork breakage) and repair by break-induced replication (BIR) and that Mus81-Mms4 and Yen1 would be redundantly required for survival (*Mayle et al., 2015*). Notably, however, also in the Flp-nick system, we observed that M phase restriction of Mus81-Mms4 allowed survival similar to WT cells (*Figure 3E–F*, note the *yen1Δ* background). In contrast, the S$^{low}$-Mus81-Mms4 construct led to pronounced sensitivity similar to the *MUS81* deletion, while S$^{high}$-Mus81-Mms4 led to an intermediary phenotype (*Figure 3E–F*). Collectively, these data show that restriction of Mus81-Mms4 to S phase renders cells sensitive to various forms of replication stress, while restriction of Mus81-Mms4 to M phase does not cause a discernible phenotype, suggesting that in budding yeast the dominant function of Mus81-Mms4 in response to replication stress is post-replicative.

## Mus81-Mms4 act as a post-replicative resolvase

To reveal the temporal control underlying the activity of Mus81-Mms4 in resolving recombination and/or replication structures, we turned to single cell cycle experiments. First, we used a single-cell-cycle setup to show that even after DNA damage induction, and recovery, the cell cycle restriction of Mus81-Mms4 expression to S or M phase remains intact (*Figure 4—figure supplement 1*). Consistent with the fact that in budding yeast cyclin-CDK complexes are unlikely to be directly regulated by DNA damage signals (*Zegerman and Diffley, 2009*), we observed restriction to S or M phase as expected. When we next treated cells with MMS in S phase and measured cell survival, we obtained a similar picture as in experiments with chronic exposure: M phase-restricted Mus81-Mms4 showed sensitivity similar to WT cells (*Figure 4A*). In contrast, cells expressing S phase-restricted Mus81-Mms4 showed hypersensitivity, with S$^{low}$-Mus81-Mms4 cells similar to the *mus81Δ* knock-out and S$^{high}$-Mus81-Mms4 displaying slightly better survival (*Figure 4A*). To have a physical read-out of repair, we used pulsed-field gel electrophoresis (PFGE), which resolves linear chromosomes, but not in the presence of replication and recombination structures. Intriguingly, *mus81Δ* deficiency has been shown to interfere with recovery of linear chromosomes after MMS treatment in S phase (*Ho et al., 2010*, see *Figure 4B–C*), but it has been unclear whether this represents a direct function of Mus81 at stalled replication forks or rather a post-replicative function in resolving recombination intermediates. When we released cells from replication fork stalling with MMS in S phase, we found that cells expressing M phase-restricted Mus81-Mms4 versions recover linear chromosomes as WT cells (*Figure 4B–C*). In contrast, cells restricting Mus81-Mms4 to S phase were strongly delayed in the appearance of resolved, linear chromosomes, as were *mus81Δ* cells (*Figure 4B–C*). These data

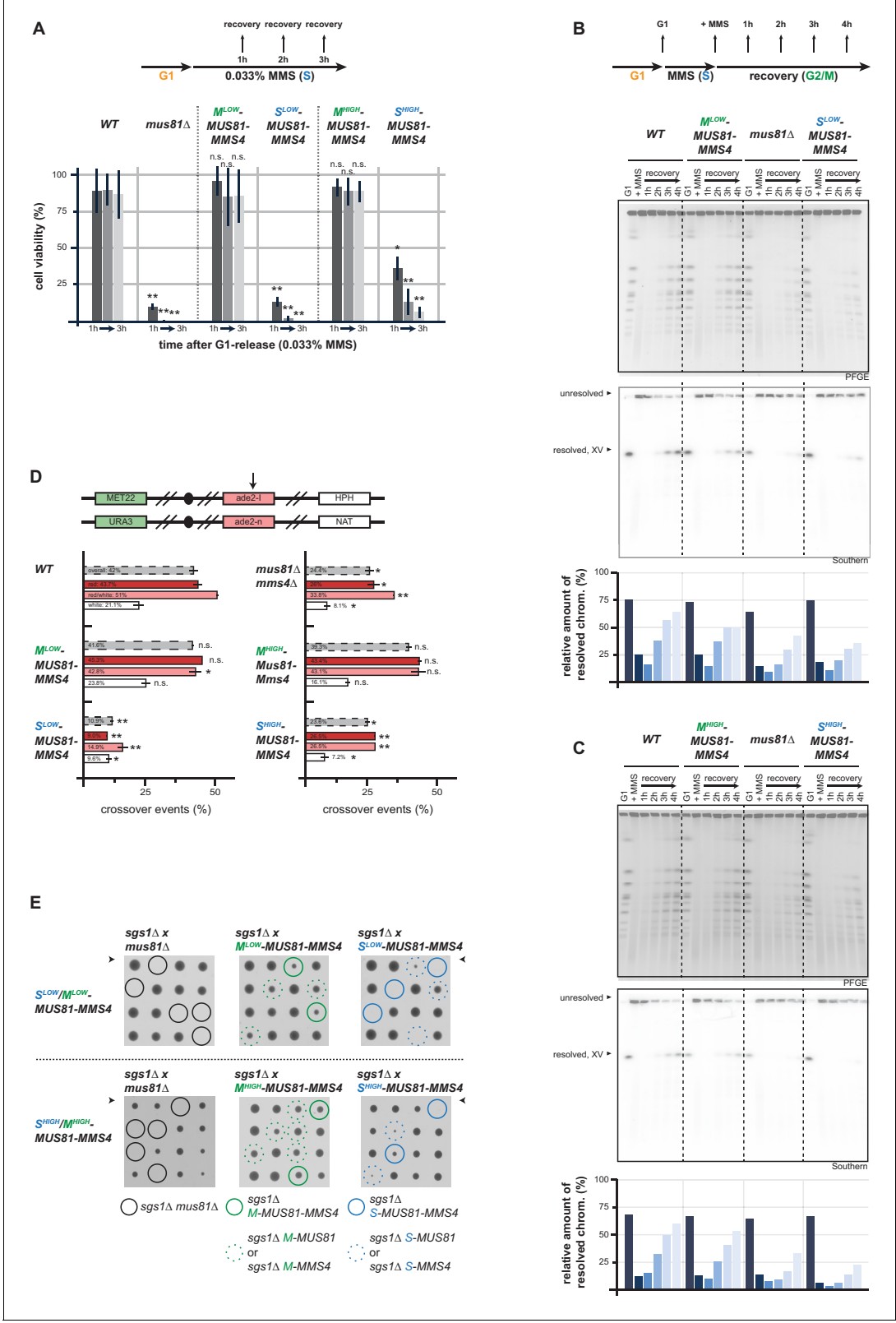

**Figure 4.** Mus81-Mms4 act as a post-replicative resolvase. (**A**) Viability after a pulse of MMS in S phase and subsequent replication fork stalling depends on the M phase function of Mus81-Mms4. Viability assay scoring survivors after pulses of MMS in S phase for one to three hours (upper). Cell viability (%) was determined by colony forming units normalized to untreated cells (0 hr) and is depicted as mean of biological replicates (n = 3) with error bars indicating standard deviation. Significance: n.s. p>0.05, *p<0.05, **p<0.005 as calculated by an unpaired Student´s T-test (see *Figure 4—*

*Figure 4 continued on next page*

*Figure 4 continued*

*source data 1* for underlying values and exact p-values). (B/C) Resolution of replication/repair intermediates arising in response to replication stalling in S phase requires mitotic Mus81-Mms4 function. PFGE analysis of cells recovering (1–4 hr in Nocodazole) from a pulse of MMS (0.033%, 1 hr) in S phase (see upper panel for experimental setup). PFGE gels were stained with EtBr or subjected to southern blot hybridization with a probe against the *ADE2* locus located on chromosome XV. The relative number of resolved chromosomes XV from the southern blots was quantified using ImageJ and is depicted below. (B) PFGE/southern analysis of M^low^-Mus81-Mms4/S^low^-Mus81-Mms4 cells. (C) PFGE/southern analysis of M^high^-Mus81-Mms4/S^high^-Mus81-Mms4 cells. (D) HR repair resulting in crossovers depends on the mitotic function of Mus81. I-SceI induced recombination assay between heterologous *ade2* alleles in diploid cells as described in *Ho et al., 2010*. Upper panel: arrangement of marker genes on chromosomes IV used for classifying the genetic outcomes of DSB repair. The arrow indicates the I-SceI site. Bottom panel: genetic outcome of repair, with overall crossover events (grey) and crossovers among individual classes (red, red/white, white) that differ in conversion tract length. Depicted are mean values from two independent experiments each scoring 400–600 cells with the standard deviation as error bars. Significance: n.s. p>0.05, *p<0.05, **p<0.005 compared to WT cells by unpaired Student´s T-test (see *Figure 4—source data 2* for underlying values and exact p-values). (E) The essential requirement of Mus81 in the absence of *SGS1*-dependent dissolution occurs during M phase. Tetrad analysis of yeast diploid cells with indicated genotypes reveals synthetic lethality between *sgs1Δ* and S^low^-/S^high^-Mus81-Mms4 while M^low^-/M^high^-Mus81-Mms4 shows no discernible effect on cell growth in the background of *sgs1Δ* (see *Figure 4—figure supplement 2* for a growth analysis of the individual spores of the tetrad analysis with S^high^-/M^high^-Mus81-Mms4).

The online version of this article includes the following source data and figure supplement(s) for figure 4:

**Source data 1.** Average, stdv and p-values of normalized colony numbers from replicates 1–3 depicted in *Figure 4A*.
**Source data 2.** Average, stdv and p-values of CO rates from two independent experiments depicted in *Figure 4D*.
**Figure supplement 1.** Cell cycle tags restrict efficiently to S or M phase also after DNA damage treatment.
**Figure supplement 2.** Mus81 function during M phase is required in the absence of Sgs1 function.

therefore indicate that Mus81-Mms4 resolves replication or recombination structures to linear chromosomes in a post-replicative manner during M phase.

To further test if the dominant M phase function of Mus81 is that of a post-replicative resolvase, we turned to DSB repair. Specifically, we used a genetic system to study the repair products of an I-SceI induced DSB in diploid cells and score for rates by which recombination intermediates are processed by resolution enzymes generating crossovers (*Ho et al., 2010*). Cells lacking Mus81-Mms4 showed a strong reduction in the formation of crossover repair products (*Ho et al., 2010*; *Figure 4D*). Notably, cells expressing M phase-restricted versions of Mus81-Mms4 were proficient in crossover formation as WT cells, while S phase-restricted versions of Mus81-Mms4 as well as the *MUS81* deletion showed reduced rates of crossover formation (*Figure 4D*). This shows that the M phase function of Mus81-Mms4 is linked to its role in forming crossovers, suggesting that in budding yeast a major function of Mus81-Mms4 is that of a post-replicative resolvase.

A large proportion of recombination intermediates is typically processed by the Sgs1-Top3-Rmi1 helicase-decatenase complex (*Gangloff et al., 1994*; *Fabre et al., 2002*; *Wu and Hickson, 2003*; *Cejka et al., 2010*). A hallmark phenotype of *mus81Δ* mutants therefore is the synthetic lethality with *sgs1Δ* (*Kaliraman et al., 2001*; *Mullen et al., 2001*; *Fabre et al., 2002*; *Bastin-Shanower et al., 2003*). A strong synthetic phenotype was observed when we restricted Mus81 expression to S phase, but no such defect was seen when we restricted Mus81 expression to M phase (*Figure 4E*, *Figure 4—figure supplement 2*). Overall, the cell cycle tag methodology therefore makes a strong case for Mus81-Mms4 having its dominant function in response to replication perturbation by acting post-replicatively in M phase, likely as a resolvase processing HR or replication intermediates. Such an M function is consistent with M phase specific phosphorylation and stimulation (*Matos et al., 2011*; *Gallo-Fernández et al., 2012*; *Matos et al., 2013*; *Saugar et al., 2013*; *Szakal and Branzei, 2013*; *Gritenaite et al., 2014*; *Princz et al., 2017*), but does not exclude an S phase function outside of the tested phenotypes.

## Premature activation of Yen1 from S to early M phase interferes with the response to replication stalling lesions

Why are resolvases cell cycle regulated? We and others have reasoned that high levels of resolvase activity during S phase may interfere with replication and the response to replication stalling (*Szakal and Branzei, 2013*; *Blanco et al., 2014*; *Duda et al., 2016*; *Pfander and Matos, 2017*). We realized that cell cycle tags could also be used to reinstall cell cycle regulation to deregulated versions of proteins, in this case resolvases. To this end we turned to a second resolvase – Yen1, which

is cell cycle-controlled and restricted to late M phase by inhibitory cyclin-CDK phosphorylation (*Kosugi et al., 2009*; *Matos et al., 2011*; *Blanco et al., 2014*; *Eissler et al., 2014*; *García-Luis et al., 2014*). A mutant version of Yen1 called *YEN1-ON* is deficient in inhibitory CDK phosphorylation sites, constitutively active throughout the cell cycle and detrimental to cellular survival after genotoxic insults as well as during meiosis (*Blanco et al., 2014*; *Eissler et al., 2014*; *Arter et al., 2018*).

To restrict Yen1-ON to specific cell cycle phases and reveal at which cell cycle phase the detrimental effects of *YEN1-ON* manifest, we combined this allele with cell cycle tags. Specifically, we used our proposed cell cycle tag workflow (*Figure 1D*) and generated two sets of G1-S-M triples expressed at low and high levels, with similar peak expression levels within each set. Tagging of Yen1 at the N-terminus interfered with protein function and we therefore constructed C-terminal cell cycle-tagged versions of Yen1-ON (*Figure 5A*, *Figure 5—figure supplement 1*). When we screened different cell cycle tag constructs, we found that Clb6$^{pClb6 \, -80bp}$-tagged, Clb2$^{pClb2 \, -60bp}$-tagged and Sic1$^{pSic1 \, u(ATA)AUG}$-tagged versions of Yen1-ON showed peak expression levels in S, M and G1 phase that were between 1.3 to 1.5-fold of Yen1-ON expressed from endogenous promoter (*Figure 5A*, *Figure 5—figure supplement 2A,C*) and will therefore be referred to as M$^{low}$-Yen1-ON, S$^{low}$-Yen1-ON and G1$^{low}$-Yen1-ON, respectively. Furthermore, Clb2$^{pClb1}$-tagged, Clb6$^{pClb6}$-tagged and Sic1$^{pSic1 \, -20bp}$-tagged versions of Yen1-ON showed 2.5 to 3-fold peak expression levels compared to endogenous Yen1-ON expressed from its endogenous promoter, but similar among the different constructs (*Figure 5A*, *Figure 5—figure supplement 2B,C*), and will be referred to as M$^{high}$-Yen1-ON, S$^{high}$-Yen1-ON and G1$^{high}$-Yen1-ON, respectively. S$^{low}$-Yen1-ON and S$^{high}$-Yen1-ON expression peaked in S phase (20–50 min and 20–60 min after G1 release, respectively), M$^{low}$-Yen1-ON and M$^{high}$-Yen1-ON expression in M phase (50–70 min) and G1$^{low}$-Yen1-ON and G1$^{high}$-Yen1-ON started to express in late M phase (70 min, *Figure 5A*), thereby confirming cell cycle restriction of Yen1-ON expression to the expected cell cycle phases.

With cell cycle-restricted versions of Yen1-ON at hand we tested known Yen1-ON phenotypes. Deregulated Yen1-ON is able to complement phenotypes of the *MUS81* deletion mutant such as MMS hypersensitivity, suggesting that both resolvases display a degree of functional redundancy (*Blanco et al., 2014*). Notably, only the M phase-restricted version of Yen1-ON and the constitutively expressed Yen1-ON were able to rescue the MMS sensitivity of cells lacking *MUS81*, while the S phase-restricted and the G1 phase-restricted versions of Yen1-ON were unable to do so (*Figure 5B*). These data further indicate that early M phase is a window of opportunity, during which a Mus81-like resolvase must act and that Yen1-ON can take this function, if specifically activated during this time.

Conversely, *YEN1-ON* itself induces hypersensitivity towards MMS compared to WT cells (*Blanco et al., 2014*; *Figure 5C–D*). We therefore tested the cell cycle-restricted versions of Yen1-ON for MMS hypersensitivity and found that restriction of Yen1-ON expression to late M and G1 using the G1$^{high}$-Yen1-ON or G1$^{low}$-Yen1-ON construct did not yield hypersensitivity, no matter whether cells were chronically exposed to MMS (*Figure 5C*) or treated with a pulse of MMS during S phase (*Figure 5D*). This suggests that restricting Yen1-ON expression to those cell cycle phases where the protein would normally be in its dephosphorylated form is sufficient to suppress the MMS hypersensitivity phenotype. In contrast, Yen1-ON expression in S or M phase caused MMS hypersensitivity that was similar to what was observed with unrestricted expression of Yen1-ON (*Figure 5C–D*). Notably, Yen1-ON phenotypes depend strongly and in a dose-dependent manner on its expression levels (*Blanco et al., 2014*; MG Blanco, personal communication). Consistently, we saw slightly increased MMS hypersensitivity in S$^{high}$-Yen1-ON and M$^{high}$-Yen1-ON compared to S$^{low}$-Yen1-ON and M$^{low}$-Yen1-ON (*Figure 5C–D*).

Therefore, we conclude that (i) the presence of deregulated Yen1 in S phase is detrimental to the cellular response to replication stalling and that (ii) deregulated Yen1 in M phase is detrimental as well but can at the same time partially compensate for the absence of Mus81-Mms4. Overall, the cell cycle tag approach therefore demonstrated that the dominant functions of Mus81-Mms4 and Yen1 manifest in those cell cycle phases – M phase and late M/G1 phase, respectively – where the proteins also become stimulated by PTM modification/demodification.

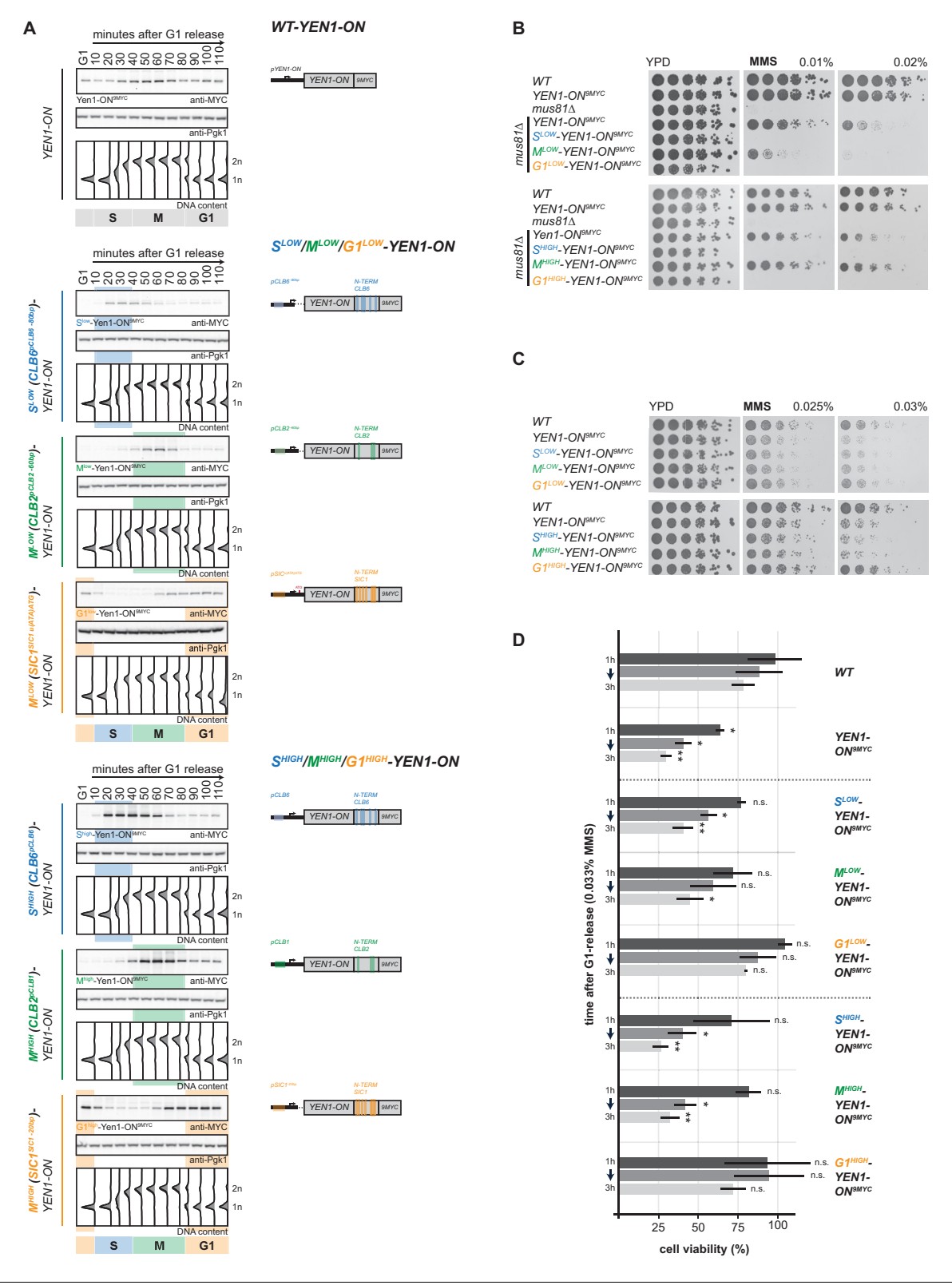

**Figure 5.** Premature activation of Yen1 in S or early M phase interferes with the response to replication stalling lesions. (**A**) Cell cycle-tagged Yen1-ON-9MYC constructs restrict expression of constitutively active Yen1-ON to S, M or G1 phase. (Left) Western blot analysis of strains expressing WT, S (S$^{low}$ (Clb6$^{pClb6\,-80bp}$)-Yen1-ON$^{9MYC}$/S$^{high}$ (Clb6$^{pClb6}$)-Yen1-ON$^{9MYC}$), M (M$^{low}$ (Clb2$^{pClb2\,-60bp}$)-Yen1-ON$^{9MYC}$/M$^{high}$ (Clb2$^{pClb1}$)-Yen1-ON$^{9MYC}$) and G1 (G1$^{low}$ (Sic1$^{pSic1\,u(ATA)ATG}$)-Yen1-ON$^{9MYC}$/G1$^{high}$ (Sic1$^{pSic1\,-20bp}$)-Yen1-ON$^{9MYC}$) phase-restricted Yen1-ON during synchronous cell cycle progression as in

*Figure 5 continued on next page*

*Figure 5 continued*

*Figure 2A*. (see *Figure 5—figure supplement 2* for quantification of peak expression levels for individual constructs). (Right) schematic representation of endogenously expressed and cell cycle-tagged Yen1-ON constructs. Blue, green and orange bars indicate location of cell cycle regulatory elements in the promoter and N-terminal degron sequence. (**B**), The M phase function of Yen1-ON is able to bypass Mus81 requirement after MMS induced replication fork stalling. Strains with indicated genotypes were chronically exposed to MMS as in *Figure 3A* (note the *mus81Δ* background). (**C–D**), Viability after MMS induced replication fork stalling decreases when Yen1-ON is restricted to S or early M phase. (**C**) Survival of indicated strains after chronic MMS exposure as in (**B**). (**D**) Viability assay after a single pulse of MMS in S phase was measured for indicated strains as in *Figure 4A* (see *Figure 5—source data 1* for underlying values and exact p-values).

The online version of this article includes the following source data and figure supplement(s) for figure 5:

**Source data 1.** average, stdv and p-values of normalized colony numbers from replicates 1–3 depicted in *Figure 5D*.
**Figure supplement 1.** Strategy for C-terminal cell cycle tagging of Yen1-ON.
**Figure supplement 2.** Analysis of 'low' and 'high' S-, M- and G1-tagged Yen1-ON peak expression levels.

## Discussion

### An advanced toolbox of cell cycle tags

Cell cycle tags are a straightforward method to restrict protein expression to specific cell cycle phases. The toolbox of constructs presented here allows straightforward introduction of cell cycle tags at the locus of interest within the budding yeast genome using standard recombination-based techniques (*Knop et al., 1999*; *Janke et al., 2004*). Importantly, with constructs that vary in peak expression, our cell cycle tag toolbox allows for the first time to restrict protein expression to different cell cycle phases and at similar peak expression levels. Similar peak expression levels are necessary, if one wants to compare phenotypes arising from restricting protein expression to different cell cycle phases or tries to unravel at which cell cycle phase a protein exhibits its essential function. To allow titration of expression levels, our cell cycle tag toolbox currently contains a total of 46 cell cycle tag constructs, with the upper expression limit being determined by cyclin promoters (*Figure 1*). So far, we have used these constructs to restrict expression of 13 proteins (*Figures 2–4*; JB and BP unpublished data). Based on this experience, we suggest the following experimental workflow: (i) Compare peak expression levels of cell cycle-restricted constructs to endogenously expressed protein. Here, a western blot against 3FLAG-tagged protein might be used (see *Pfander and Diffley, 2011* for pYM-3FLAG tagging constructs). (ii) Due to the dynamic expression regime, a given cell cycle-tagged construct is unlikely to give endogenous expression levels throughout the cell cycle phase of interest. This problem is further aggravated if experimental conditions are varied (for example: growth on liquid vs solid media, cell cycle arrest, or drug treatment activating cell cycle checkpoints). Therefore, we suggest finding triples of S-, M- and G1-tag constructs (or sometimes pairs) with similar peak expression levels. Furthermore, we advise to select two sets of triples/pairs (called 'low' and 'high' throughout the manuscript), which vary in peak expression levels and thereby allow to separate phenotypes arising from under- or overexpression from those arising from cell cycle restriction. Although expression may vary for individual proteins, combinations of constructs that regularly gave us similar peak expression levels are Clb6$^{pClb6}$ (or Clb6$^{pClb5}$), Clb2$^{pClb1}$, Sic1$^{pSic1\ -20bp}$ for the high expressing set and Clb6$^{pClb6\ -80bp}$, Clb2$^{pClb2-60bp}$ (or Clb2$^{pClb1\ -150bp}$), Sic1$^{pSic1\ u(ATA)ATG}$ for the low set. (iii) Genetics will often indicate whether N-terminal tagging is compatible with protein function as such, but an additional control for the functionality of the tagged constructs is desirable. For Mus81-Mms4 we used cleavage of DNA junctions in vitro (*Figure 2D*). Collectively these considerations are expected to allow interpretation of the cell cycle restriction experiment and reveal cell cycle stage-specific functionality of the protein of interest.

### The essential function of the structure-selective nuclease Mus81-Mms4 manifests in M phase

Phenotypic analysis suggests that Mus81-Mms4 plays a major role in the response to replication stalling (*Xiao et al., 1998*; *Interthal and Heyer, 2000*; *Boddy et al., 2001*; *Mullen et al., 2001*; *Doe et al., 2002*; *Bastin-Shanower et al., 2003*; *Kai et al., 2005*; *Saugar et al., 2013*). As such, it is seemingly contradictory that Mus81-Mms4 becomes post-translationally modified and stimulated in its activity only after S phase, when cells enter M phase (*Matos et al., 2011*; *Matos et al., 2013*; *Gallo-Fernández et al., 2012*; *Saugar et al., 2013*; *Szakal and Branzei, 2013*; *Princz et al., 2017*).

Mutations abolishing Mus81-Mms4 phosphorylation showed pronounced phenotypes and M phase-specific stimulation of Mus81-Mms4 shows hallmark signs of switch-like activation (*Matos et al., 2011*; *Gallo-Fernández et al., 2012*; *Matos et al., 2013*; *Szakal and Branzei, 2013*; *Princz et al., 2017*). Nonetheless, a possible S phase function as well as the relative contribution of S and M phase phenotypes have been widely discussed (*Xiao et al., 1998*; *Boddy et al., 2000*; *Interthal and Heyer, 2000*; *Haber and Heyer, 2001*; *Kaliraman et al., 2001*; *Mullen et al., 2001*; *Doe et al., 2002*; *Fabre et al., 2002*; *Bastin-Shanower et al., 2003*; *Kai et al., 2005*; *Hanada et al., 2007*; *Shimura et al., 2008*; *Regairaz et al., 2011*; *Fu et al., 2015*; *Xing et al., 2015*; *Lemaçon et al., 2017*). In this study, we have applied the cell cycle tag approach to tackle this question and observed that restriction of Mus81-Mms4 expression to S phase induces strong phenotypes similar to those of the *MUS81* deletion, while restriction of Mus81-Mms4 to M phase allows full functionality. The essential function(s) of Mus81-Mms4 therefore appear to be specific to M phase, correlating with its M phase-specific stimulation.

Consequently, this raises the question about the mechanism underlying the Mus81-Mms4 M phase function. We think that our data are generally consistent with a model whereby Mus81-Mms4 acts as a resolvase that processes an HR intermediate (for example a HJ or a D loop (*Boddy et al., 2001*; *Kaliraman et al., 2001*; *Chen et al., 2001*; *Constantinou et al., 2002*; *Doe et al., 2002*; *Bastin-Shanower et al., 2003*; *Ciccia et al., 2003*; *Gaillard et al., 2003*; *Osman et al., 2003*; *Whitby et al., 2003*; *Fricke et al., 2005*; *Ehmsen and Heyer, 2008*; *Taylor and McGowan, 2008*; *Schwartz et al., 2012*; *Pepe and West, 2014b*)). This HR intermediate could originate from the usage of HR (or template switch replication) to bypass replication stalling lesions or DNA breaks arising in S phase (*Liberi et al., 2005*; *Branzei et al., 2008*; *Giannattasio et al., 2014*; *Branzei and Szakal, 2016*). As such, the prime function of Mus81-Mms4 would be to aid resolution of sister chromatids (*Roseaulin et al., 2008*; *Wechsler et al., 2011*; *Wyatt et al., 2013*; *Mankouri et al., 2013*; *Matos et al., 2013*; *Szakal and Branzei, 2013*; *Gritenaite et al., 2014*; *Princz et al., 2017*) and share functional overlap with STR-dependent dissolution (*Gangloff et al., 1994*; *Fabre et al., 2002*; *Wu and Hickson, 2003*; *Ira et al., 2003*).

Mus81-Mms4 might also have more than one M phase function. For example, it is possible that Mus81-Mms4 might act on replication forks that persist until M phase, similar to what has been shown for MUS81 in human cells, which acts in the MIDAS pathway to process replication forks at sites of under-replicated DNA and to thereby initiate repair by BIR (*Minocherhomji et al., 2015*; *Duda et al., 2016*). Furthermore, Mus81-Mms4 may also act at a later step in BIR to switch from an extending D-loop mechanism of DNA synthesis to a type of DNA synthesis that is more similar to canonical DNA replication, a model that has been suggested by a study in budding yeast (*Mayle et al., 2015*). Interestingly, we observed using the same experimental set-up as Mayle et al. that M phase-restricted Mus81-Mms4 is entirely sufficient for survival after induction of the Flp-nickase (*Figure 3E,F*), suggesting that a possible function of Mus81-Mms4 in BIR takes place in M phase thus bearing a similar cell cycle profile as MIDAS (*Minocherhomji et al., 2015*). In line with our conclusions, and while this manuscript was in preparation, the Pasero and Aguilera labs published a preprint showing by several approaches a post-replicative function of Mus81-Mms4 in response to replication blockage (*Pardo et al., 2019*).

We emphasize that our study can by no means rule out that Mus81-Mms4 also functions in S phase. However, at least in budding yeast we are not aware of any data in the current literature that would necessitate to evoke such an S phase function for Mus81-Mms4. This raises the question of whether budding yeast Mus81-Mms4 can serve as a paradigm for other species. The fact that human MUS81 complexes are cell cycle regulated (*Wyatt et al., 2013*; *Duda et al., 2016*) suggests that Mus81-Mms4 with its essential M phase function may indeed be a good model for the Mus81 biology in other organisms. However, we caution that additional regulatory mechanisms exist in other systems. Fission yeast Mus81-Eme1 harbours for example an additional layer of control by the DNA damage checkpoint (*Boddy et al., 2000*; *Kai et al., 2005*; *Froget et al., 2008*; *Dehé et al., 2013*). Human MUS81 has two accessory subunits (EME1 and EME2; *Ciccia et al., 2003*), which are likely differentially regulated (*Matos et al., 2011*; *Duda et al., 2016*). Indeed, there is overwhelming genetic data showing that in human cells MUS81 is required for the cellular response to replication perturbation (*Hanada et al., 2007*; *Shimura et al., 2008*; *Regairaz et al., 2011*; *Fugger et al., 2013*; *Naim et al., 2013*; *Ying et al., 2013*; *Pepe and West, 2014a*; *Xing et al., 2015*; *Fu et al., 2015*; *Minocherhomji et al., 2015*; *Lemaçon et al., 2017*). While some of these data could be

explained by a role for MUS81 as a post-replicative resolvase – similar to what is shown here for the budding yeast protein – an earlier role at stalled replication forks in S phase cannot be excluded. Perhaps the strongest case for an S phase function of human MUS81 is made by several genetic studies showing that cells depleted for MUS81 display perturbed DNA replication, a constitutive DNA damage response and defects in the response to replication perturbation (*Hanada et al., 2007*; *Shimura et al., 2008*; *Regairaz et al., 2011*; *Buisson et al., 2015*; *Fu et al., 2015*; *Xing et al., 2015*). These phenotypes could arise from a direct function at replication forks in S phase (perhaps in processing/cleavage of stalled replication forks), which would then be absent from the budding yeast system (or so far elusive). Since at this stage a possible indirect effect upon MUS81 deletion or depletion can also not be excluded, we suggest that methodologies analogous to the cell cycle tag system described here might be useful to ascertain the relative contribution of M and S phase functions in human cells.

### Premature resolvase activation is detrimental to the response to replication stalling

A second and so far under-utilized application of cell cycle tags is to install de novo cell cycle regulation on a deregulated mutant protein. We demonstrated this strategy using the deregulated *YEN1-ON* allele of the Yen1 resolvase (*Blanco et al., 2014*). Specifically, we showed that the previously described MMS hypersensitivity phenotype of *YEN1-ON* cells is generated by premature activation of Yen1 in S and early M phase. These data corroborate the importance of restricting post-translational activation of Yen1 specifically to late M and G1 phase. They also argue against Yen1 having a function in S phase and raise the question about what detrimental effects premature resolvase activation might be causing.

Interestingly, these data suggest that differences between the Mus81-Mms4- and Yen1-dependent mechanisms exist and fit to both enzymes having distinct temporal activation profiles. Differential activation of Mus81-Mms4 and Yen1 could perhaps simply serve the purpose to equip cells with at least one highly active SSE at different cell cycle stages from early M to the G1/S transition (*Wild and Matos, 2016*). Furthermore, temporal activation establishes hierarchy in the corresponding resolution/dissolution mechanisms. In particular, we favour a three-tiered hierarchy for enzymes that process recombination intermediates: first, STR-dependent dissolution is fully active in S phase (*Ashton et al., 2011*; *Versini et al., 2003*; *Bizard and Hickson, 2014*; *Grigaitis et al., 2020*), second, Mus81-Mms4 gets stimulated in early M and third, Yen1 would become activated only in late M and as a measure of last resort. Such a hierarchy could be a means to counteract cross-overs and loss-of-heterozygosity in mitotically dividing cells (*Ho et al., 2010*; *Matos et al., 2013*; *Szakal and Branzei, 2013*; *Blanco et al., 2014*), but it may simply reflect differential efficiency of competing molecular mechanisms. Furthermore, it is possible that the three mechanisms are directed towards distinct substrates. In order to identify the exact nature of these substrates, molecular genetic assays will be necessary and we suggest they be carried out with very precise genetic perturbation provided by cell cycle tag methodology.

## Materials and methods

**Key resources table**

| Reagent type (species) or resource | Designation | Source or reference | Identifiers | Additional information |
|---|---|---|---|---|
| Antibody | anti-FLAG (rabbit-polyclonal) | Sigma | Cat# F7425, RRID:AB_439687 | WB (1:2000) |
| Antibody | anti-FLAG M2-Peroxidase (mouse-monoclonal) | Sigma | Cat# A8592, RRID:AB_439702 | WB (1:3000) |
| Antibody | anti-MYC (mouse-monoclonal) | Millipore | Cat# 05–724, clone 4A6, RRID:AB_11211891 | WB (1:2000) |

*Continued on next page*

*Continued*

| Reagent type (species) or resource | Designation | Source or reference | Identifiers | Additional information |
|---|---|---|---|---|
| Antibody | anti-Clb2 (rabbit-polyclonal) | Santa Cruz | Cat# sc-9071, RRID:AB_667962 | WB (1:500) |
| Antibody | anti-Hsp70 (mouse-polyclonal) | Enzo Life Sciences | Cat# ADI-SPA-822, RRID:AB_10615940 | WB (1:10000) |
| Antibody | anti-H4 (rabbit-polyclonal) | Abcam | Cat# ab10158, RRID:AB_296888 | WB (1:2000) |
| Antibody | anti-Nsp1 (mouse-monoclonal) | Abcam | Cat# ab4641, RRID:AB_304549 | WB (1:10000) |
| Strain, strain background (*S. cerevisiae*) | strain background, W303 | *Rothstein, 1983* | | |
| Strain, strain background (*S. cerevisiae*) | yeast strains | This study | | *Supplementary File 1* |
| Recombinant DNA reagent | plasmid backbone, pYM-N31 | *Janke et al., 2004* Euroscarf | P30284 | |
| Recombinant DNA reagent | plasmid backbone, pRS303 | ATCC | Cat# 77138 | |
| Recombinant DNA reagent | plasmids | This study | | *Supplementary file 1* |
| Sequence-based reagent | S1 | *Janke et al., 2004* | PCR primers | cgtacgctgcaggtcgac |
| Sequence-based reagent | S4 | *Janke et al., 2004* | PCR primers | catcgatgaattctctgtcg |
| Sequence-based reagent | yeGFP S1 | This study | PCR primers | attgtaatacgactcact atagggcgaattggag ctccaccgcggtggcg gccgccgtacgctgca ggtcgac |
| Sequence-based reagent | yeGFP S4 | This study | PCR primers | ctaattcaaccaaaattg ggacaacaccagtgaa taattcttcacctttagaca tcatcgatgaattctctgtcg |
| Sequence-based reagent | Mus81 S1 | This study | PCR primers | caaagtttcaaaggatt gatacgaacacacattc ctagcatgaaagcatgc gtacgctgcaggtcgac |
| Sequence-based reagent | Mus81 S4 | This study | PCR primers | caactaattcttgtaaccatt caatatataggtcttttaagtt tgatgagagttccatcgatg aattctctgtcg |
| Sequence-based reagent | Mms4 S1 | This study | PCR primers | acaatgtatggattatgg tatagaataatagtagtc acatattgcagctagttaa cgtacgctgcaggtcgac |
| Sequence-based reagent | Mms4 S4 | This study | PCR primers | gaatactggcatcgtttct tgaatctttgtcctcaaca aaatcaacgatctggctc atcgatgaattctctgtcg |
| Commercial assay or kit | In-Fusion HD Cloning | Clontech | 639648 | |

*Continued on next page*

*Continued*

| Reagent type (species) or resource | Designation | Source or reference | Identifiers | Additional information |
|---|---|---|---|---|
| Chemical compound, drug | Sytox-green | Invitrogen | S7020 | |
| Chemical compound, drug | RNaseA | Sigma Aldrich | R4875 | |
| Chemical compound, drug | ProteinaseK | Sigma Aldrich | P2308 | |
| Chemical compound, drug | Pierce ECL Western Blotting Substrate | Thermo Fisher | 2106 | |
| Chemical compound, drug | Nocodazol | Sigma Aldrich | M1404 | |
| Chemical compound, drug | Hydroxyurea | Sigma Aldrich | H8627 | |
| Chemical compound, drug | MMS | Sigma Aldrich | 129925 | |
| Chemical compound, drug | Camptothecin | Sigma Aldrich | C9911 | |
| Peptide, recombinant protein | alpha-Factor | MPIB core facility | | |
| Other | Zymolyase Z100T | Roth | 9329.2 | |
| Other | Pulsed Field certified Agarose | BioRad | 1620138 | |
| Other | Amersham Protran Premium 0.45 um Nitrocellulose membrane | GE Healthcare | 10600003 | |
| Other | Amersham Hybond N+ Nylon membrane | GE Healthcare | RPN203B | |
| Other | NuPAGE Novex, 4–12% BIS-TRIS gels | Invitrogen | NP0323 | |
| Software, algorithm | T-Test calculator | GraphPad | GraphPad, RRID:SCR_000306 | http://www.graphpad.com/quickcalcs/ttest2 |

## Yeast strains

All yeast strains are based on W303 (*Rothstein, 1983*) and constructed by genetic crossing and transformation techniques. ORF deletion as well as N-terminal cell cycle tagging was done using standard techniques (*Knop et al., 1999*; *Janke et al., 2004*) and is described in more detail in the paragraph construction of cell cycle-tagged strains. A list of all yeast strains used in this study can be found in *Supplementary file 1* – Table 1.

## Construction of cell cycle-tagged strains

Tagging constructs for N-terminal cell cycle tagging of genes are based on the pYM-N plasmid (*Janke et al., 2004*) and harbour the regulatory sequences of the corresponding cyclin (promoter + N-terminus) together with a 3FLAG-tag and the NAT marker sequence flanked by S1 and S4 primer

sequences (*Figure 1—figure supplement 1*). Plasmid constructs were generated using standard molecular biology techniques and truncation of the 5′UTR or insertion of upstream out of frame ATGs was achieved by site-directed mutagenesis techniques. A list of tagging constructs for N-terminal cell cycle tagging of genes can be found in *Supplementary file 1* - Table 2.

Amplification of the N-terminal tagging cassettes was achieved by PCR using S1 and S4 primer sequences (see Key Resources table for sequences) fused to a 55 bp sequence homologous to the promoter-gene junction of the corresponding gene (*Figure 1—figure supplement 1*) (see Key Resources table for sequences of S1, S4 primers coupled to the GFP, Mus81 and Mms4 promoter-gene junction sequences). The PCR product was transformed into competent yeast cells and correct integration of the tagging constructs was verified by genotyping PCR using two primer pairs, whereby the first tested integration of the tagging construct and the second verified deletion of the endogenous promoter sequence. Expression of the gene fusion product was then verified by western blotting (see *Figure 1—figure supplement 1* for a protocol of the experimental workflow).

For C-terminal cell cycle tagging of Yen1-ON, constructs were assembled together with the gene of interest within the integrative pRS303 vector backbone, linearized by restriction enzyme cutting and integrated into the HIS3 locus. Correct integration of the plasmids was checked by genotyping PCR using two primer pairs verifying integration of a single copy of the plasmid and expression of the tagged protein was verified by western blotting (see *Figure 5—figure supplement 1* for a protocol of the workflow).

## Antibodies

Detection of proteins was achieved by using antibodies listed in the Key Resources table.

## DNA content measurement

Cell cycle progression was analysed by DNA content measurements by flow cytometry. $1 \times 10^7 - 2 \times 10^7$ cells were harvested and resuspended in 1 ml of fixation buffer (70% ethanol + 50 mM Tris pH 8). Cells were washed 1x with 1 ml of 50 mM Tris pH 8 and incubated in 520 µl of RNase solution (500 µl 50 mM Tris pH 8 + 20 µl RNase A (10 mg/ml in 10 mM Tris pH 7.5, 10 mM MgCl2) over night at 37˚C. Next, cells were treated with 220 µl proteinase K solution (200 µl 50 mM Tris pH 8 + 20 µl proteinase K (10 mg/ml in 50°C glycerol, 10 mM Tris pH 7.5, 25 mM CaCl2) for 30 min at 50˚C. Afterwards, cells were resuspended in 500 µl 50 mM Tris pH 8, sonicated (5′′, 50% Cycle, minimum power) and stained in SYTOX solution (1:1000 in Tris pH 8). Fluorescence intensity was measured at 520 nm using MACSquant Analyzer 10 (Milteny Biotech) and the data was analysed using FlowJo (FlowJo, LLC).

## Acrylamide gel electrophoresis and western blotting

Separation of proteins was achieved using standard SDS-polyacrylamid gel electrophoresis in 4–12% Novex NuPage BisTris precast gels (ThermoFisher) with MOPS buffer (50 mM MOPS, 50 mM Tris-base, 0.1% SDS, 1.025 mM EDTA, adjusted to pH 7.7). Afterwards, proteins were transferred to nitrocellulose membrane (Amersham Protran Premium 0.45 µm NC) by wet blotting in western transfer buffer (48 mM Tris-base, 39 mM glycine, 0.0375% SDS, 20% methanol). Membranes were incubated with the primary antibodies (diluted at concentrations indicated in the Key resources table in milk buffer: 2.5% milk powder, 0.5% BSA, 0.5 % NP-40, 0.1% Tween-20, 50 mM Tris pH 7.5, 137 mM NaCl, 3 mM KaCl) over night at 4˚C or at room temperature for 2 hr when using mouse-anti-FLAG directly coupled to HRP. Appropriate secondary antibodies coupled to HRP were applied at room temperature for 2 hr. Washing of the membranes was performed three times for 5 min with western wash buffer (50 mM Tris pH 7.5, 137 mM NaCl, 3 mM KaCl, 0.2% NP-40) and incubated with Pierce ECL western blotting substrate (ThermoFisher) according to the manufacturer's instruction. Chemiluminescence was detected using a tabletop film processor (OPTMAX, Protec) or with an iBright FL1000 imaging system (ThermoFisher).

## Preparation of whole cell extracts (alkaline lysis/TCA)

$2 \times 10^7$ cells were resuspended in 1 ml of pre-cooled water and mixed with 150 µl of freshly prepared lysis solution (1.85 M NaOH, 7.5% beta-mercaptoethanol). Lysis was performed at 4˚C for 15 min and protein precipitation was achieved by adding 150 µl of 55% pre-cooled TCA solution and

incubation for 10 min. After centrifugation and aspiration of the supernatant, protein pellets were resuspended in 50 µl HU-buffer (8 M Urea, 5% SDS, 200 mM Tris pH 6.8, 1.5% dithiothreitol, traces of bromophenol blue) and incubated at 65°C for 10 min.

## Synchronization of cells, cell cycle release, viability

Generally, cell cycle arrests and releases were performed as described in *Reusswig et al., 2016*. Cells were grown to log-phase ($OD_{600}$0.4–0.6) in YP + 2% glucose (YPD) prior to arrest. To arrest cells in G1, S and M phase, the cultures were supplemented with α-factor (5 µg/ml, MPIB), nocodazole (5 µg/ml, Sigma-Aldrich) and hydroxyurea (HU) (200 mM, Sigma-Aldrich) for 2 hr, respectively.

Release from G1 was performed by washing cells twice with prewarmed YP followed by resuspending cells in the same volume of prewarmed YPD. To ensure that cells run through the cell cycle only once, α-factor (5 µg/ml) was added back to the cultures 40 min after release from G1. For DNA damage treatment in S phase and analysis of fully replicated chromosomes by PFGE cells were released from G1 into prewarmed YPD containing 0.033% MMS for 1 hr. Afterwards, cells were washed twice again with prewarmed YP and resuspended in prewarmed YPD containing nocodazole (5 µg/ml).

To determine survival of MMS treatment cells were kept in MMS containing medium (0.033%) and plated in triplicates onto YPD plates at time points indicated in the figures. Colony-forming units were counted after incubation at 30°C for 3 days. Viability experiments were performed in three independent biological replicates and the standard deviations of those experiments are represented as error bars in the corresponding bar charts. Statistical significance for the viability of individual strains compared to the wild-type was calculated using an unpaired Student´s T-test. These calculations were done using the GraphPad web-tool 'T-test calculator' (http://www.graphpad.com/quickcalcs/ttest2).

For subsequent analysis of DNA or protein content, aliquots (1 $OD_{600}$, approx. $1 \times 10^7$ cells) were withdrawn from the culture at indicated time points. For flow cytometric analysis, cells were resuspended in fixation buffer (70% ethanol + 50 mM Tris pH 8) and incubated at 4°C for at least 30 min prior to further processing (see section DNA content measurement). For western analysis, cells were snap-frozen in liquid nitrogen and stored at −80°C prior to further processing (see section alkaline lysis/TCA).

## Chronic treatment with genotoxic agents

To assess viability of yeast strains on MMS, CPT and HU containing solid medium (prepared 1 day prior), cells from stationary grown over-night cultures were spotted with a starting concentration of $OD_{600}$ 0.5 in serial dilution (1:5) and incubated at 30°C for 2 days. All spottings were done in 2–3 biological replicates, each containing two technical replicates.

## Pulsed-field gel electrophoresis (PFGE) and southern blotting

Cells were fixed and embedded in agarose plugs as described in *Finn and Li, 2013*. Plugs were loaded on a 1% (w/v) agarose (Pulsed-field certified, BioRad) gel in 0.5 x TBE (45 mM Tris, 45 mM borate, 0.5 mM EDTA). Electrophoresis was carried out in 14°C cold 0.5 x TBE in a CHEF DR-III system (initial switch time 60 s, final switch time 120 s, 6 V/cm, angle 120°C, 24 hr). Afterwards, the gel was stained with 1 µg/ml ethidium-bromide in 0.5 x TBE for 1 hr and destained with deionized water. Images were taken using a VWR GenoSmart gel documentation system.

For southern blotting the DNA was nicked in 0.125 M HCl for 10 min, denatured in 1.5 M NaCl, 0.5M NaOH for 30 min and neutralized by 0.5 M Tris, 1.5 M NaCl (pH 7.5) for 30 min. The DNA was transferred onto a Hybond-N+ membrane (GE healthcare) and cross-linked with UV-light (Stratagen, auto-crosslink function). The membrane was probed with a radioactive (α−32P dCTP) labelled ADE2 fragment and imaged using a Typhoon FLA 9000 imaging system.

## DSB-induced recombination assay

The DSB-induced recombination assay was performed as described previously (*Ho et al., 2010*). In brief, diploid cells were grown to log-phase ($OD_{600}$0.4–0.6) in liquid YPAR (YP + 40 mg/l adenine + 2% raffinose). DSB formation (I-SceI expression) was induced by adding galactose (final concentration 2%) to the cultures for 1.5 hr. Afterwards cells were plated onto YPAD (YPD + 10 mg/l adenine),

incubated for 3–4 days and replica plated onto YPAD +Hyg +Nat, YPAD + Hyg, YPAD + Nat, SC -Ura, SC -Met and SCR -ADE +Gal to classify recombination events. The different classes evaluated arise from repair of DSBs by either short tract or long tract gene conversion which produces ade2-n or ADE+ recombinants, respectively (white class: two short tract conversions; red class: two long tract conversions; red/white class: on long tract, on short tract conversion). CO events in the different classes were measured by the number of colonies that have rendered both daughter cells homozygous for the HPH and NAT marker. In each experiment 400–600 cells per strain were evaluated for the individual class of repair and the experiment was independently repeated twice. Standard deviations were calculated and included as error bars. Statistical significance for the CO rates of individual strains and classes compared to the wild-type was calculated using an unpaired Student´s T-test. These calculations were done using the GraphPad web-tool 'T-test calculator' (http://www.graph-pad.com/quickcalcs/ttest2).

## Mus81-Mms4 nHJ cleavage assay

Asynchronous mitotic cultures were generated by inoculating 1 l of YP-Raffinose (20 g/l bactopeptone, 10 g/l yeast extract, 20 g/l D-(+)-Raffinose) with overnight cultures, to an $OD_{600}$ of ~0.2. After cells have grown to an exponential stage ($OD_{600}$ ~0.5) Mus81-Mms4 expression was induced by adding 20 g/l D-(+)-Galactose to the culture. The cells were then grown for 2 hr and harvested by centrifugation.

For Myc-affinity purifications yeast pellets were resuspended in 200 µl of lysis buffer (40 mM Tris-HCl (pH 7.5 at 25°C), 150 mM NaCl, 20 mM β-glycerolphosphate, 1 mM NaF, 0.5 mM EGTA, 0.1 mM EDTA, 15% (V/V) glycerol, 0.1% (V/V) NP-40, 1 mM DTT, 2 mM PMSF, 1x Complete Protease inhibitor cocktail (Roche)) and lysed with glass beads. Obtained lysates were cleared by centrifugation and normalized. Myc-tagged Mus81-Mms4 was then immunoprecipitated using mouse monoclonal antibodies to Myc (9E10) coupled to agarose beads (AminoLink Plus), pre-blocked with 1 mg/ml BSA in lysis buffer. Immunoprecipitations were done on a rotating wheel for 1 hr at 4°C. Prior western blotting or DNA cleavage assays the beads were extensively washed with the wash buffer (40 mM Tris-HCl (pH 7.5 at 25°C), 150 mM NaCl, 15% (V/V) glycerol, 0.1% (V/V) NP-40) and dephosphorylated by treating the beads with bacteriophage λ protein phosphatase (New England Biolabs) in 1x protein metallophosphatases (PMP) buffer supplemented with 1 mM $MnCl_2$. Reactions were assembled on ice and incubated at 30°C for 15 min. The dephosphorylated Mus81-Mms4 complexes were then subjected to SDS-PAGE gel and used for DNA cleavage assays.

DNA cleavage assays on beads were adapted from previously described protocols (*Grigaitis et al., 2018*; *Matos and West, 2017*). In short, 20 µl of reaction mixture (20 mM Tris-Ac, pH 7.5 (25°C), 3 mM $MgAc_2$, 1 mM DTT, 100 µg/ml BSA) containing 5 nM fluorescently labeled nicked Holliday junction, prepared as previously described (*Grigaitis et al., 2018*) were added to dry aspirated anti-Myc beads, corresponding to ~20 µl of volume. Reactions were assembled on ice and initiated by transferring them to 30°C. Reactions were performed for 2 hr at 30°C, shaking 800 rpm, stopped by the addition of 5x STOP solution (100 mM Tris-Ac, (pH 7.5 at 25°C), 50 mM EDTA, 2.5% (m/w) SDS, 10 mg/ml Proteinase K) and incubated for 1 hr at 37°C, shaking 800 rpm. Subsequently 5 µl of 6x DNA loading dye (13 mM Tris-HCl (pH 8.0 at 25°C), 40 mM EDTA, 0.32% (V/V) SDS, 250 µM Ficoll 400, 0.4 µM OrangeG) were added. Reaction mixtures were analysed on a native 10% polyacrylamide gel in 1x TBE (89 mM Tris-borate (pH 8.4 at 25°C), 2 mM EDTA) by running the electrophoresis for 1 hr 15 min at 7.5 V/cm. The gel was then imaged with a Typhoon scanner (GE Healtcare).

## Nuclear fractionation

The protocol for preparation of nuclear fractions was adapted from *Pasero et al., 1999*. In brief, 30 ml of the synchronized cultures (at different time points after G1 release) were fixed with sodium azide (total concentration 0.1%) and harvested by centrifugation (2 min, 3500 rpm, RT). Cells were resuspended in 5 ml of 100 mM EDTA-KOH, pH 8, 10 mM DTT and incubated with constant shaking at 30°C for 10 min. Afterwards, cells were resuspended in 2 ml of YPD/1.1 M sorbitol and 0.5 mg/ml Z100T was added (incubation for 20 min, constant shaking, 30°C) and subsequently recovered in 2 ml YPD/1.1 M sorbitol + 0.5 mM PMSF (incubation for 10 min, constant shaking, 30°C). Zymolyase digested cells were resuspended in 2 ml of ice cold breakage buffer (10 mM Tris-HCl, pH 7.4, 40

mM KCl, 4 mM EDTA-KOH, 0.25 mM Spermidine, 0.1 mM Spermine, 18% Ficoll, 1% beta-mercap-toethanol, 1% aprotinin, 0.5 mM PMSF, 300 µg/ml benzamidine, 1 µg/ml pepstatin A, 0.5 µg/ml leupeptin) and dounced on ice using a tight pestle (about 30 strokes) to lyse cells efficiently (Total extract sample). Lysis was verified microscopically and extracts were centrifuged to remove unbroken cells (2 x, 12 min, 5000 g, 4°C). Finally, cleared extracts were centrifuged at high speed to separate the nuclei from the cytoplasm (15 min, 21000 g, 4°C). The supernatant (Cytoplasmic sample) was removed, nuclei washed once by rinsing the pellet with 1 ml of cold breakage buffer and the nuclear pellet was resuspended in 100 µl 0.25 x buffer A (10 mM Tris-HCl, pH 7.4, 40 mM KCl, 4 mM EDTA-KOH, 0.25 mM Spermidine, 0.1 mM Spermine) + 0.5 mM PMSF (Nuclear sample).

## Acknowledgements

We thank Uschi Schkölziger and Tobias Freimann for technical assistance, Jonathan Baxter, Lotte Bjergbaek, Gregorz Ira, Lorraine Symington and Wolfgang Zachariae for strains, constructs and protocols, Georgios I Karras for design of an early version of the Clb5-S-tag construct, Miguel G Blanco for communication of unpublished data, Miguel G Blanco, Hans Hombauer and members of the Pfander and Jentsch labs for stimulating discussion and critical reading of the manuscript. This work was supported by the Max Planck Society (to BP), by ETH Zürich and the Swiss National Science Foundation (to JM). Work in the Pfander lab is supported by grants of the Max Planck Society and by the Deutsche Forschungsgemeinschaft (DFG, German Research Foundation) – Project ID PF794/3-1; Project-ID 213249687 – SFB 1064 Deutsche Forschungsgemeinschaft (DFG, German Research Council).

## Additional information

### Funding

| Funder | Grant reference number | Author |
| --- | --- | --- |
| Schweizerischer Nationalfonds zur Förderung der Wissenschaftlichen Forschung | | Joao Matos |
| Eidgenössische Technische Hochschule Zürich | | Joao Matos |
| Max-Planck-Gesellschaft | | Boris Pfander |
| Deutsche Forschungsgemeinschaft | PF794/3-1 | Boris Pfander |
| Deutsche Forschungsgemeinschaft | 213249687 – SFB 1064 | Boris Pfander |

The funders had no role in study design, data collection and interpretation, or the decision to submit the work for publication.

### Author contributions

Julia Bittmann, Conceptualization, Formal analysis, Investigation, Methodology, Project administration, Writing - review and editing; Rokas Grigaitis, Formal analysis, Investigation, Methodology; Lorenzo Galanti, Formal analysis, Investigation, Methodology, Writing - review and editing; Silas Amarell, Investigation, Methodology; Florian Wilfling, Methodology; Joao Matos, Resources, Formal analysis, Supervision, Writing - review and editing; Boris Pfander, Conceptualization, Resources, Formal analysis, Supervision, Funding acquisition, Investigation, Methodology, Writing - original draft, Writing - review and editing

### Author ORCIDs

Julia Bittmann (iD) https://orcid.org/0000-0001-6527-7383
Lorenzo Galanti (iD) http://orcid.org/0000-0003-2538-3581

Joao Matos [ID] http://orcid.org/0000-0002-3754-3709
Boris Pfander [ID] https://orcid.org/0000-0003-2180-5054

**Decision letter and Author response**
Decision letter https://doi.org/10.7554/eLife.52459.sa1
Author response https://doi.org/10.7554/eLife.52459.sa2

## Additional files

### Supplementary files

• Supplementary file 1. Yeast strains and Plasmids. Table 1. Yeast strains used in this study. Table 2. Cell cycle tagging plasmids used in this study Table 3. Other plasmids used in this study

• Transparent reporting form

### Data availability

All data generated or analysed during this study are included in the manuscript and supporting files.

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
