## [Decision Letter]

**Acceptance summary:**

This paper is based on the development of a sophisticated strategy to turn on and off the expression of proteins at specific stages of the cell cycle in yeast. Using this approach the authors show that the activity of structure-specific nucleases is required after S phase.

**Decision letter after peer review:**

Thank you for submitting your article "An advanced cell cycle tag toolbox reveals principles underlying temporal control of structure-selective nucleases" for consideration by *eLife*. Your article has been reviewed by three peer reviewers, including Bernard de Massy as the Reviewing Editor and Reviewer #1, and the evaluation has been overseen by a Reviewing Editor and Kevin Struhl as the Senior Editor. The following individuals involved in review of your submission have agreed to reveal their identity: Pierre-Henri Gaillard (Reviewer #2).

The reviewers have discussed the reviews with one another and the Reviewing Editor has drafted this decision to help you prepare a revised submission.

Summary:

In this paper by Bittmann et al., an advanced cell cycle tag toolbox reveals principles underlying temporal control of structure-selective nucleases, and a powerful strategy is developed to analyze the requirement of structure selective endonucleases, in particular, Mus81-Mms4 during the cell cycle in *S. cerevisiae*. This is achieved by inducing the expression of these genes by cell cycle-dependent promoters and by inactivating the proteins by cell cycle-dependent degradation (degrons). By combining the use of various cyclin promoters, N-terminal cyclin degrons and truncations of the 5'-UTR of their constructs, they have generated a set of 42 cell cycle tags that not only often allow for a tighter restraint of the production of a protein to G1, S or M-phase than previously developed cell-cycle tag strategies, but also allow for some degree of control on protein expression levels.

The authors conclude that the expression of Mus81-Mms4 in M-phase but not in S phase is required for function (resolution of DNA damage). This is mainly a confirmation, with several functional assays, of previous studies showing the stimulation of Mus81-Mms4 in late G2/M. Mus81-Mms4 activity is thus concluded to be post-replicative and not to be required during S phase.

This is an impressive piece of work and the manuscript is overall clearly written with data that are, for the most part, robust and rather convincing.

However, several aspects of the paper have to be revised and improved in order to provide convincing interpretations and conclusions, and thus to present data going beyond the confirmation and validation of previously established cell-cycle regulation of these activities.

Essential revisions:

1) The authors have used constructs with high and low expression levels for Mus81-Mms4. As they discuss in the paper (subsection “An advanced toolbox of cell cycle tags”), having an endogenous expression level is important. As shown in Figure 2 S_high_ and M_high_ constructs have expression levels higher than endogenous whereas S_low_ and M_low_ have expression levels closer to endogenous. Therefore, all results should be based on S_low_ and M_low_ constructs. Experiments with these constructs should be presented in the main figures (including the assays presented in Figure 3 and Figure 4) and should be the core of the discussion for phenotype analysis and comparison with the *mus81Δ* strain. S_high_ and M_high_ could be included in supplementary material.

Note that in subsection “Cell cycle-restriction of Mus81-Mms4 reveals its essential function during M phase”, with 4-5 fold for Mus81 and 2-3 fold for Mms4 higher expression levels, use of "highly" and "moderately", respectively, would be more appropriate than "slightly". How was this quantified? No explanation is provided in Figure 2—figure supplement 1A.

In relation to this issue, how is phosphorylated Mms4 quantified? The 2-3 fold higher expression level seems by eye may be under-estimated, compared to the 4-5 fold higher expression level of Mus81. It would be worth loading gels of phosphatase treated samples to facilitate the quantification of single bands rather than of smears.

2) Throughout the manuscript, the authors write about the "activation" of Mus81-Mms4 in late G2 and M-phase. This is misleading since it suggests that the enzyme is inactive in other phases of the cell-cycle. The use of "up-regulation" would seem more appropriate.

3) The question of the role of Mus81-Mms4 in S phase is a central issue in this paper, several revisions should be made: In a few experiments, the M_high_-Mus81-Mms4 construct does not fully rescue the phenotype of a Mus81-deletion. This suggests that M phase restriction is limiting the function of this complex. Other interpretation could be the effect of the tags. In addition, some phenotypes are partially suppressed by S_high_ constructs. Here, again at least two alternative interpretations: either Mus81-Mms4 has an S phase function, or its expression after S, explains the phenotype. This question is expected to be solved with the use of S_low_ constructs however.

If the expression of S_low_ construct still partially suppresses phenotypes, the authors should test the expression of an S phase restricted construct of Mms4(8A). An S phase function would not be expected to be sensitive to the alteration of phospho-sites.

The authors should test lower concentrations of CPT and MMS with the S_low_ and M_low_ constructs as this might allow revealing an S-phase contribution of Mus81-Mms4.

In relation to these questions, the author should explain and provide experimental interpretation for the lack of suppression by M_low_ at high MMS and CPT concentration (Figure 2D).

4) In terms of presenting the data and presenting conclusions, some revisions should be made.

First for data presentation:

– In the text, the authors oversimplify in a biased manner the results of the data of some of the experiments and the text does not match the data on the figures:

– Subsection “Cell cycle-restriction of Mus81-Mms4 reveals its essential function during M phase”, "*mus81Δ* were hypersensitive to all agents": this does not really show for HU.

– Subsection “Cell cycle-restriction of Mus81-Mms4 reveals its essential function during M phase”, "Restricting Mus81-Mms4 to S phase generated hypersensitivity that was similar to what we observed in *mus81Δ"*: this is not correct.

– Please check other figures and text for the description of data.

For interpretations:

– The authors do acknowledge that an S phase function of Mus81-Mms4 cannot be excluded, but the overall tone leaves the reader with the idea that it only acts in M-phase. This also applies to the Abstract, where the Mus81-Mms4 is concluded to act as a post-replicative resolvase. There are a couple of studies (Fu et al., 2015; Xing et al., 2015) that show that MUS81 is required for optimal replication dynamics in human cells (assessed by fiber analysis). It is difficult to imagine how MUS81 could contribute to replication fork progression measured in DNA combing experiments if it only acts in mitosis. The authors should discuss this to leave the door open for possible functions of Mus81-Mms4 in S-phase that they have not identified with their experimental setup. The implication in *S. cerevisiae* in BIR (Mayle et al., 2014) also suggests activity on replicating chromatids. If the authors want to conclude definitively on this issue, they have to provide a molecular assay for Mus81-Mms4 activity with a kinetic study to follow in parallel cell cycle progression.

5) A major issue is that although the authors show very clearly how the proteins of interest are expressed under the various cell-specific constructs (Figure 2B and Figure 2—figure supplement 2B), they do not show the expression in the conditions used for functional assays, i.e. after MMS and CPT treatment. The authors have to show that expression is as expected after MMS and CPT, for S_low_ and M_low_ constructs.

[Editors' note: further revisions were suggested prior to acceptance, as described below.]

Thank you for submitting your article "An advanced cell cycle tag toolbox reveals principles underlying temporal control of structure-selective nucleases" for consideration by *eLife*. Your article has been reviewed and the evaluation has been overseen by a Reviewing Editor and Kevin Struhl as the Senior Editor.

We would like to draw your attention to changes in our revision policy that we have made in response to COVID-19 (https://elifesciences.org/articles/57162). Specifically, we are asking editors to accept without delay manuscripts, like yours, that they judge can stand as *eLife* papers without additional data, even if they feel that they would make the manuscript stronger. Thus, the revisions requested below only address clarity and presentation.

We thank the authors for the extensive revisions and additional work. The result is an impressive, convincing and high-quality paper.

We agree with the removal of Cdc5 experiments which as the authors note appears to add additional unknown to the analysis. The presentation is quite clear, and we understand the interest of providing both High and Low constructs for assaying the various phenotypes.

One slight modification seems to be important:

The molecular assays in Figure 4B and C are an important step in the analysis. Visually, one can see differences in band intensities, but it is really impossible to evaluate the significance without the quantification that measures the ratio of the two bands. This quantification should not be as Figure 4—figure supplement 2 but should be inserted in the main Figure 4. Because space is limited, one option is to crop the southern blot to allow for space for the graph (and to put the full size southern in a figure supplement). It is also possible to show the quantified data in Figure 4 for High constructs only.

Subsection “Additional Information**”**: should be Figure 5—source data 1 (instead of Figure 4—source data 1).

---

## [Author Response]

Essential revisions:1) The authors have used constructs with high and low expression levels for Mus81-Mms4. As they discuss in the paper (subsection “An advanced toolbox of cell cycle tags”), having an endogenous expression level is important. As shown in Figure 2 S_high_ and M_high_ constructs have expression levels higher than endogenous whereas S_low_ and M_low_ have expression levels closer to endogenous. Therefore, all results should be based on S_low_ and M_low_ constructs. Experiments with these constructs should be presented in the main figures (including the assays presented in Figure 3 and Figure 4) and should be the core of the discussion for phenotype analysis and comparison with the mus81Δ strain. S_high_ and M_high_ could be included in supplementary material.Note that in subsection “Cell cycle-restriction of Mus81-Mms4 reveals its essential function during M phase”, with 4-5 fold for Mus81 and 2-3 fold for Mms4 higher expression levels, use of "highly" and "moderately", respectively, would be more appropriate than "slightly".How was this quantified? No explanation is provided in Figure 2—figure supplement 1A.In relation to this issue, how is phosphorylated Mms4 quantified? The 2-3 fold higher expression level seems by eye may be under-estimated, compared to the 4-5 fold higher expression level of Mus81. It would be worth loading gels of phosphatase treated samples to facilitate the quantification of single bands rather than of smears.

We thank the reviewers for highlighting the issue of expression levels. Indeed, the major technological aim of our study was to improve the cell cycle tag methodology by developing a toolbox of cell cycle tag constructs that allows titrating expression levels. Moreover, we aimed to not only give researchers experimental tools at hand, but also a workflow they can apply when studying the function of their protein of interest. Expression levels and their influence on phenotypes have been largely disregarded in previous studies and this may have led to misinterpretation. We therefore hope that our paper will establish a new gold standard for future cell cycle tag studies.

Specifically, we would like to point out that it is important to consider two aspects when “replacing” a continuously (constantly) expressed protein with a dynamically expressed, cell cycle-restricted protein. The first is – as the reviewers point out – the peak expression level, the second aspect is the dynamicity of expression (“How quickly does protein expression rise?”). We fear that the initial presentation of our manuscript has led the reviewers to focus very much on peak expression levels, while the dynamic aspect has received less attention. We apologize for this misunderstanding and have therefore included an additional theoretical paragraph (subsection “An advanced toolbox of cell cycle tags”) on our strategy for cell cycle-tag experiments (Combination of “High” and “Low” expressing tags (see below and Figure 1D)) and additional data to validate our approach (Figure 2B-C).

Essentially, our strategy is based on the assumption that with current tools it is impossible to generate a version of your protein of interest that altogether (a) is tightly cell cycle restricted, (b) has a peak expression similar to endogenous levels and (c) is constantly expressed throughout a specific cell cycle phase. This assumption was verified by the following experimental observations (see also our response to point #3).

i) We observed that constructs with peak expression levels comparable to that of the endogenous protein (we call those “Low” constructs throughout the manuscript) showed underexpression at cell cycle transitions. For example, M_low_-Mus81-Mms4 which peaks in expression in mid M phase (50-60 min after G1 release, Figure 2A), is underexpressed in early M phase (35-45 min after G1 release, new Figure 2B) and this results in a shortened window during which we can detect hyperphosphorylated Mus81-Mms4 in M phase (new Figure 2C).

ii) We conduct additional experiments (as detailed below in the response to point #3) to verify that the small DNA damage hypersensitivity of M_low_-Mus81-Mms4 expressing cells (which is not observed in M_high_-Mus81-Mms4 expressing cells) is due to underexpression in M phase. Conversely, we show that the small difference in DNA damage sensitivity of S_high_-Mus81-Mms4 expressing cells and *mus81Δ* cells (which is not observed in S_low_-Mus81-Mms4 expressing cells) comes from slight leakage of expression of this construct into M phase.

To mitigate this deficit, we suggest a strategy that involves two sets of cell cycle tagged constructs (as shown in Figure 1D, Figure 2—figure supplement 1, Figure 5—figure supplement 2) at different expression levels. The first set (we call it “Low” throughout the manuscript) will give peak expression levels similar to endogenous expression levels and is tightly restricted, but suffers from potential “underexpression” at cell cycle transitions. The second set (we call it “High” throughout the manuscript) will give peak expression levels higher than endogenous protein levels (“overexpression”) and likely less tight restriction, but avoids “underexpression” at cell cycle transitions. Conclusions should then be based on the collective results of the two datasets, which allows to implement the influence of both peak expression levels and expression dynamics.

For this reason, and in this specific case, we do not follow the reviewers’ suggestion to base our conclusion exclusively on the “Low” set of tags. Indeed, we have undergone a major effort to be able to present all experiments side-by-side in the “High” and “Low” background for both Mus81-Mms4 and Yen1, including new genetic experiments in Figure 4A, B, D, E, Figure 4—figure supplement 1, and Figure 5B-D.

Regarding the technical sub-points:

i) We fully agree with the reviewers that the phrase “slightly overexpressed” was badly chosen, particularly as it contrasts with our “High”/“Low” nomenclature and we corrected usage throughout the manuscript.

ii) We quantified protein levels at individual timepoints using ImageJ. First, protein levels of Mus81 or Mms4 were normalized to the Pgk1 loading control. Normalized protein levels of Mus81 or Mms4 were then expressed as fold difference to the corresponding WT sample and the two analysed timepoints were averaged. We have now included a detailed description in the figure legends. In case of Mms4, we have quantified the signal over the phosphorylated and unphosphorylated protein species.

iii) We appreciate the reviewers’ concern that the presence of several bands (or even a smear) may complicate quantification. It would be indeed problematic, if we compared Mms4 levels from M phase (two bands) with Mms4 from G1 or S phase (one band). Comparing Mms4 levels between different M phase samples appears not to be associated with large inaccuracies and we verified this under gel conditions were phosphorylated and unphosphorylated species collapsed to one band (Author response image 1). Lastly, we would like to mention that we observed roughly similar trends for Mus81 and Mms4 constructs, supporting our conclusion (Figure 2—figure supplement 1A-B).

**Author response image 1. sa2fig1:** Quantification of Mus81 and Mms4 levels after resolving and non-resolving gel running conditions. Expression levels were quantified using Image-J and signals at the individual timepoints were divided by the corresponding Pgk1 sample to normalize to overall protein levels. Normalized protein levels were compared to WT expression at the corresponding timepoint (fold difference) and the resulting fold difference is depicted in the graphs below the western blots. For the mitotic Mms4 samples (M_low_-Mms4 and M_high_-Mms4) the analysis was done both after resolving (4-12% gel, 90min) and after non-resolving (12% gel, 25 min) gel conditions to show the validity of quantifying phosphorylated Mms4 signals. The Mus81 and S phase Mms4 samples were run with resolving gel conditions only. A, Quantification of peak expression levels from Figure 2—figure supplement 1, extended by an analysis with non-resolving gel conditions for the mitotic Mms4 samples. B, Quantification of peak expression levels from samples of the experiment shown in Figure 2C with resolving and non-resolving gel conditions.

2) Throughout the manuscript, the authors write about the "activation"of Mus81-Mms4 in late G2 and M-phase. This is misleading since it suggests that the enzyme is inactive in other phases of the cell-cycle. The use of "up-regulation" would seem more appropriate.

We agree with the reviewers that it is important to use correct wording in order to express what happens to Mus81-Mms4 in M phase. “Activation” could be misleading if interpreted in terms of a digital on/off model. “Upregulation” is, however, also misleading as in the life sciences it is usually interpreted as increased expression (we verified that up-regulation would lead to misconception with several “test readers” that were unfamiliar with the story). We think that “enhanced/increased/upregulated Mus81 function” would be better and we have now used this term throughout the manuscript. We define Mus81 function as its “ability to find, recognize and cleave DNA joint molecule structures” (subsection “Cell cycle-restricted expression of Mus81-Mms4”).

3) The question of the role of Mus81-Mms4 in S phase is a central issue in this paper, several revisions should be made: In a few experiments, the M_high_-Mus81-Mms4 construct does not fully rescue the phenotype of a Mus81-deletion. This suggests that M phase restriction is limiting the function of this complex. Other interpretation could be the effect of the tags. In addition, some phenotypes are partially suppressed by S_high_ constructs. Here, again at least two alternative interpretations: either Mus81-Mms4 has an S phase function, or its expression after S, explains the phenotype. This question is expected to be solved with the use of S_low_ constructs however.If the expression of S_low_ construct still partially suppresses phenotypes, the authors should test the expression of an S phase restricted construct of Mms4(8A). An S phase function would not be expected to be sensitive to the alteration of phospho-sites.The authors should test lower concentrations of CPT and MMS with the S_low_ and M_low_ constructs as this might allow revealing an S-phase contribution of Mus81-Mms4.In relation to these questions, the author should explain and provide experimental interpretation for the lack of suppression by M_low_ at high MMS and CPT concentration (Figure 2D).

To summarize, this point of criticism basically asks: does our data exclude a function of Mus81-Mms4 during S phase? During the revision we have undergone significant experimental effort to scrutinize this question. Collectively, our data does not lend support for an S phase function of Mus81-Mms4 as summarized in the following points:

i) Is M_high_-Mus81-Mms4 able to fully complement Mus81-Mms4 functions?

We thank the reviewers for this good observation, but the very slight survival defect in M_high_-Mus81-Mms4 cells treated with 0.033% MMS (Figure 3A of the previous version) could only be sporadically observed. Indeed, our survival assays (for unrelated reasons now consistently in W303 *RAD5*+ background) do not show any difference to WT (Figure 3B, Figure 4A) This is consistent with all other experiments, where M_high_-Mus81-Mms4 cells phenotypically behave like WT cells. (Figure 3B, D, F, Figure 4C-E)

ii) Is M_low_-Mus81-Mms4 able to fully complement Mus81-Mms4 functions?

A very related question to (i). Indeed, we observed in several experiments that M_low_-Mus81-Mms4 expressing cells showed reduced Mus81 function compared to WT cells (Figure 3A, Figure 4B, 4E), while in other experiments no significant defect could be observed (Figure 3C, 3E, Figure 4A, 4D). This could be for one of two reasons: (a) M_low_-Mus81-Mms4 is in some form under-expressed (see also our response to point #1, and our results with M_high_-Mus81-Mms4) or (b) having Mus81-Mms4 in M phase is not sufficient for full Mus81-Mms4 functionality, because of a putative, then missing S phase function. To address this important point, we generated a strain that harbors both M_low_-Mus81-Mms4 and S_low_-Mus81-Mms4. If the M_low_-Mus81-Mms4 survival defect was due to an S phase function of Mus81-Mms4, we would expect that this would be cured by the additional presence of Mus81-Mms4 during S phase. This, however, was not the case. The M_low_-Mus81-Mms4 + S_low_-Mus81-Mms4 cells showed the same sensitivity as M_low_-Mus81-Mms4 cells (Figure 3—figure supplement 1D,E), suggesting that the phenotype is rather caused by underexpression of functional M phase Mus81-Mms4 in M_low_-Mus81-Mms4 cells.

iii) Does S-phase restricted Mus81-Mms4 show residual function?

Indeed, we fully agree with the reviewers that S_high_-Mus81-Mms4 cells (but not S_low_-Mus81-Mms4 cells) show some residual Mus81-Mms4 function and do not phenocopy the *mus81Δ* knock-out in all experiments (see Figure 3B, lowest concentration of MMS, CPT or HU). This could be for one of two reasons: (a) there is an S phase function of Mus81-Mms4; (b) the S phase restriction of Mus81-Mms4 is imperfect in the S_high_-Mus81-Mms4 (basically too high a fraction of Mus81-Mms4 remains after S phase and then gets phosphorylated and hyperactivated in M). We favor (b) for several reasons. First, the S_low_-Mus81-Mms4 constructs phenocopy the *mus81Δ* knock-out in all experiments (also at lower MMS and CPT concentrations, Figure 3—figure supplement 1A), which we think is inconsistent with a significant contribution of S phase Mus81-Mms4 to the tested functions (Figure 3A, 3C, 3E, Figure 4A, 4B, 4D, 4E). Second, as the reviewers suggested we have now mutated all M phase-specific Mms4 phosphorylation sites (*mms4 14A*) in the S_high_-Mus81-Mms4 construct. If the residual function of the S_high_-Mus81-Mms4 construct was due to an S phase function it should not be influenced by the M phase-specific phosphorylation sites. In contrast, if the residual function was due to imperfect restriction of S_high_-Mus81-Mms4 (bleeding into M Phase), this residual function should be killed by the *mms4 14A* mutation. Indeed, S_high_-Mus81-mms4 14A phenocopied *mus81Δ* (Figure 3—figure supplement 1B,C). We therefore do not think that we should take the residual functionality observed in S_high_-Mus81-*Mms4* (but not S_low_-Mus81-Mms4) cells as an indication of an S phase function of Mus81-*Mms4*.

We therefore thank the reviewers for the suggestion to scrutinize our experiments and to explore the possibility of a Mus81-Mms4 S phase function. Our data does not provide support for such an S phase function in the phenotypes tested. This does obviously not rule out any kind of S phase function, and we hope that our discussion provides a balanced view on this point.

4) In terms of presenting the data and presenting conclusions, some revisions should be made.First for data presentation:– In the text, the authors oversimplify in a biased manner the results of the data of some of the experiments and the text does not match the data on the figures:– Subsection “Cell cycle-restriction of Mus81-Mms4 reveals its essential function during M phase”, "mus81Δ were hypersensitive to all agents": this does not really show for HU.– Subsection “Cell cycle-restriction of Mus81-Mms4 reveals its essential function during M phase”, "Restricting Mus81-Mms4 to S phase generated hypersensitivity that was similar to what we observed in mus81Δ": this is not correct.– Please check other figures and text for the description of data.For interpretations:– The authors do acknowledge that an S phase function of Mus81-Mms4 cannot be excluded, but the overall tone leaves the reader with the idea that it only acts in M-phase. This also applies to the Abstract, where the Mus81-Mms4 is concluded to act as a post-replicative resolvase. There are a couple of studies (Fu et al., 2015; Xing et al., 2015) that show that MUS81 is required for optimal replication dynamics in human cells (assessed by fiber analysis). It is difficult to imagine how MUS81 could contribute to replication fork progression measured in DNA combing experiments if it only acts in mitosis. The authors should discuss this to leave the door open for possible functions of Mus81-Mms4 in S-phase that they have not identified with their experimental setup. The implication in *S. cerevisiae* in BIR (Mayle et al., 2014) also suggests activity on replicating chromatids. If the authors want to conclude definitively on this issue, they have to provide a molecular assay for Mus81-Mms4 activity with a kinetic study to follow in parallel cell cycle progression.

We thank the reviewers for pointing out parts of the text, which may have been imprecise. We think these have been isolated cases and hope the reviewers agree with us that our manuscript has not been oversimplified in a biased manner.

Regarding data presentation:

– We changed the statement to: “*mus81Δ* cells were hypersensitive to MMS and CPT and showed reduced growth on HU containing medium (Figure 3A-B).” (subsection “Mus81-Mms4 restricted to M phase, but not S phase is sufficient for the response to genotoxic insults”).

– The statement to “S_low_-Mus81-Mms4 expressing cells showed similar phenotypes as *mus81Δ* (Figure 3A, Figure 3—figure supplement 1A).*”* (subsection “Mus81-Mms4 restricted to M phase, but not S phase is sufficient for the response to genotoxic insults”).

Regarding interpretations:

– We agree with the reviewers that it is impossible to exclude that Mus81-Mms4 could have functions outside of M phase, even though we failed to find evidence for such a function. We explicitly mention this possibility in the Discussion section and the findings regarding an S phase function in other organisms (subsection “Synchronization of cells, cell cycle release, viability”) and we have carefully revisited the manuscript text to make sure there is no misunderstanding regarding the interpretation of our data. Therefore, while we do not want to propose that Mus81-Mms4 *only* acts in M phase, what we do like to put forward, however, is that Mus81-Mms4 functions, which manifest in the well characterized phenotypes of *mus81Δ* mutants, can be explained from its activity (or lack thereof) during M phase.

Conversely, we do not know of any experimental evidence in the literature that would indicate an S phase function for budding yeast Mus81-Mms4. The BIR phenotype (shown for example in Mayle et al., 2014) can not necessarily be taken as support for an S phase function, since BIR has been shown to occur post-replicatively (discussed for example in Saada et al., 2018, see also in particular Pardo et al., 2019 (a recent preprint from the Pasero lab)).

We are of course aware of data obtained in other organisms that can be interpreted in favor of an S phase function for Mus81 orthologues. In our discussion we try to provide a balanced view of whether and in how far lessons learned from budding yeast research on Mus81-Mms4 including the present study should be extrapolated to other organisms and make the case that advanced experimental tools such as our cell cycle tag methodology are needed in other models, too.

5) A major issue is that although the authors show very clearly how the proteins of interest are expressed under the various cell-specific constructs (Figure 2B and Figure 2—figure supplement 2B), they do not show the expression in the conditions used for functional assays, i.e. after MMS and CPT treatment. The authors have to show that expression is as expected after MMS and CPT, for S_low_ and M_low_ constructs.

This is an important point, and indeed, even though several (most!) previous studies used cell cycle tags in the context of DNA repair, it had not been verified that the presence of DNA damage might not fundamentally change the behavior of the cell cycle tags. Theoretical considerations suggest that cell cycle tags should work also in the presence of DNA damage, as in budding yeast (in contrast to metazoans) DNA damage signaling does not target cyclin-CDK.

To confirm this hypothesis, we now show that cells treated with a high dose MMS pulse during synchronous progression through S phase show functional restriction of S_low_-/ S_high_-Mus81-Mms4 to S phase and of M_low_/M_high_-Mus81-Mms4 to G2/M (Figure 4—figure supplement 1, see DNA content measurement by FACS and anti-Clb2 western blots as indicators of cell cycle progression).

[Editors' note: further revisions were suggested prior to acceptance, as described below.]

One slight modification seems to be important:The molecular assays in Figure 4B and C are an important step in the analysis. Visually, one can see differences in band intensities, but it is really impossible to evaluate the significance without the quantification that measures the ratio of the two bands. This quantification should not be as Figure 4—figure supplement 2 but should be inserted in the main Figure 4. Because space is limited, one option is to crop the southern blot to allow for space for the graph (and to put the full size southern in a figure supplement). It is also possible to show the quantified data in Figure 4 for High constructs only.

We have now included the quantification graphs in Figure 4 and removed it from Figure 4—figure supplement 2.

Subsection “Additional Information**”**: should be Figure 5—source data 1 (instead of Figure 4—source data 1).

Thank you, we have introduced the change.